# LSD1 protects against hippocampal and cortical neurodegeneration

Michael A. Christopher[1,2], Dexter A. Myrick[1,2], Benjamin G. Barwick (iD) [2,3], Amanda K. Engstrom[1,2], Kirsten A. Porter-Stransky[4], Jeremy M. Boss[3], David Weinshenker[4], Allan I. Levey[5,6] & David J. Katz[1]

To investigate the mechanisms that maintain differentiated cells, here we inducibly delete the histone demethylase LSD1/KDM1A in adult mice. Loss of LSD1 leads to paralysis, along with widespread hippocampus and cortex neurodegeneration, and learning and memory defects. We focus on the hippocampus neuronal cell death, as well as the potential link between LSD1 and human neurodegenerative disease and find that loss of LSD1 induces transcription changes in common neurodegeneration pathways, along with the re-activation of stem cell genes, in the degenerating hippocampus. These data implicate LSD1 in the prevention of neurodegeneration via the inhibition of inappropriate transcription. Surprisingly, we also find that transcriptional changes in the hippocampus are similar to Alzheimer's disease (AD) and frontotemporal dementia (FTD) cases, and LSD1 is specifically mislocalized to pathological protein aggregates in these cases. These data raise the possibility that pathological aggregation could compromise the function of LSD1 in AD and FTD.

[1] Department of Cell Biology, Emory University School of Medicine, Atlanta, GA 30322, USA. [2] Graduate Division of Biological and Biomedical Science, Emory University, Atlanta, GA 30322, USA. [3] Department of Microbiology and Immunology, Emory University School of Medicine, Atlanta, GA 30322, USA. [4] Department of Human Genetics, Emory University School of Medicine, Atlanta, GA 30322, USA. [5] Department of Neurology, Emory University School of Medicine, Atlanta, GA 30322, USA. [6] Emory University Alzheimer's Disease Research Center, Emory University School of Medicine, Atlanta, GA 30322, USA. Michael A. Christopher and Dexter A. Myrick contributed equally to this work. Correspondence and requests for materials should be addressed to D.J.K. (email: djkatz@emory.edu)

L SD1/KDM1a (hereafter referred to as LSD1) is an amine oxidase histone demethylase. In conjunction with the CoREST complex, it specifically demethylates mono-methylation and di-methylation of lysine 4 on histone H3 (H3K4me1/2), but not H3K4me3[1, 2]. Alternatively, when associated with the androgen receptor complex, LSD1 has been shown to demethylate H3K9me2[3]. LSD1 homozygous mutant mice arrest at embryonic day 5.5 and fail to properly elongate the egg cylinder, before being resorbed by embryonic day 7.5[4, 5]. In addition, loss of LSD1 results in olfactory receptor choice[6] and circadian rhythm defects[7] when conditionally deleted in mice, along with defects in plasma cell[8] and hematopoietic differentiation[9] in vitro, and pituitary[4], hematopoietic stem cell[10] and trophoblast stem cell[11] differentiation defects in vivo. These defects, along with developmental phenotypes in yeast[8], Arabidopsis[12], Drosophila[13, 14], and Caenorhabditis elegans[15], indicate that LSD1 may function during changes in cell fate. For example, in mouse embryonic stem cells (ES cells), LSD1 binds to the promoter and enhancers of the critical stem cell genes, Oct4, Sox2, Klf4, and Myc[16]. Upon differentiation, LSD1 is required to remove H3K4me1 to repress the transcription of these stem cell genes and enable proper ES cell differentiation[16]. Similarly, LSD1 has also been implicated in regulating stem cell gene transcription during the differentiation of hematopoietic stem cells[10].

Although LSD1 has many roles throughout development, little is known about its function in differentiated cells. However, one hint comes from studies of the LSD1-containing CoREST complex, which has been implicated in the maintenance of cell fate by repressing the transcription of neuronal genes in non-neuronal cell types[17, 18]. Based on this finding, we hypothesized that LSD1 may function similarly in the maintenance of other differentiated cell types. To address this possibility, we inducibly deleted Lsd1 in adult mice. Loss of LSD1 leads to paralysis, along with widespread neuronal cell death in the hippocampus and cortex, and associated learning and memory deficits. Here we have chosen to focus on the function of LSD1 in preventing hippocampus neurodegeneration, and the potential link to human neurodegenerative disease. In the degenerating hippocampus, we detect transcriptional changes in pathways implicated in human neurodegeneration. This suggests that LSD1 may prevent neuronal cell death by repressing common neurodegenerative pathways. In the degenerating neurons, we also detect the inappropriate expression of stem cell genes. This indicates that LSD1 may be part of an epigenetic maintenance program that continuously prevents inappropriate transcription. Surprisingly, we also find that LSD1 mislocalizes with pathological aggregates specifically in Alzheimer's disease (AD) and frontotemporal dementia (FTD) cases, and the genome-wide transcriptional changes in the degenerating Lsd1 hippocampus specifically correlate with those found in AD and FTD cases. These data raise the possibility that LSD1 function could be affected in these dementias.

## Results

**LSD1 is continuously required to prevent neurodegeneration.** To determine whether LSD1 is required in terminally differentiated cells within the brain, we inducibly deleted Lsd1 in adult mice by crossing floxed Lsd1 mice[4, 6, 19–21] to the Cagg-Cre tamoxifen inducible Cre transgene[22–26] (hereafter referred to as Lsd1[CAGG]). LSD1 is expressed widely in the mouse brain. Specifically, immunofluorescence detected LSD1 protein in the nuclei of NeuN positive neurons throughout the brain, including the hippocampus and cerebral cortex (Supplementary Fig. 1a–l). LSD1 protein is also present in astrocytes (Supplementary Fig. 2a–d, i–l) and oligodendrocytes (Supplementary Fig. 3a–d, i–l, q–t), but not microglia (Supplementary Fig. 4a–h). Tamoxifen injection in Lsd1[CAGG] animals resulted in the widespread loss of

LSD1 protein in hippocampal and cerebral cortex neurons between 4 and 9 weeks after the final injection (Fig. 1a–d). However, surprisingly, at this time point LSD1 protein remained unchanged in astrocytes (Supplementary Fig. 2e–h, m–p) and oligodendrocytes (Supplementary Fig. 3e–h, m–p, u–x) throughout the brain. Thus, within the brain, LSD1 loss is confined to neurons. As a result, Lsd1[CAGG] animals enable us to interrogate the result of losing LSD1 specifically in these neurons.

We do not observe any defects in non-tamoxifen-injected Cre positive Lsd1[CAGG] mice, nor in tamoxifen-injected Cre minus Lsd1[CAGG] littermate controls (hereafter used as controls in all subsequent experiments). However, all (n = 45) tamoxifen-injected Lsd1[CAGG] mice developed a severe motor deficit between 4 and 9 weeks after deletion, characterized initially by weakness in the hindlimbs followed by weakness in the forelimbs. These deficits are associated with hindlimb clasping, failure to maintain body posture, docile behavior, an inability to keep eyes open and ultimately, death (Fig. 1e–g, and Supplementary Movie 1 showing the terminal phenotype used in subsequent assays). Development of this motor defect occurred rapidly, with generally 1 week elapsing between initial onset and full defect. Importantly, the full motor defect occurred within 4–9 weeks after tamoxifen injection regardless of age at Lsd1 deletion (Fig. 1g). This suggests that LSD1 is required throughout adulthood to protect against the development of these deficits. Though both males and females ultimately exhibit the motor defect, the number of days after tamoxifen injection to reach the terminal motor phenotype was longer in males compared to females (Fig. 1g inset). It is unclear at the moment why there is a small sex specific difference in the timing of this defect.

To investigate this phenotype further, we examined the spinal cords, neuromuscular junctions, muscles, and brains of Lsd1[CAGG] mice. Mutant spinal cords appeared morphologically normal and the number of motor neurons in the spinal cord did not significantly differ from control littermates (Supplementary Fig. 5a, b). We also did not detect any defects in the morphology of neuromuscular junctions, or in myelination of the spinal cord (Supplementary Fig. 5c–f). Upon examination of limb muscles, we observed severe atrophy in the soleus muscle, as indicated by the much smaller diameter of the muscle cells, and moderate atrophy of the tibialis anterior muscle (Supplementary Fig. 5g–j). However, we did not find any evidence of muscle degeneration, suggesting the motor defect is not due to complications in muscles.

Although, we do not detect degeneration in the spinal cord or hindlimb muscle, we find widespread severe neurodegeneration in the hippocampus and cerebral cortex of Lsd1[CAGG] mice (Fig. 1h, i). As a result, we have initially focused here on the function of LSD1 in preventing this neurodegeneration. Within the hippocampus, many neuronal nuclei of the CA1, CA3, dentate gyrus, and cerebral cortex were pyknotic, and displayed a corresponding loss of the dendrite marker MAP2, as well as the axon marker Tau (Fig. 1j–s). Of these hippocampal regions, the CA1 was the most affected with 77.3 ± 5.2% pyknotic nuclei (average with s.e.m.), while the CA2 and CA3 were moderately affected (Supplementary Fig. 6a, b). Within individuals, the percent of condensed nuclei in all regions of the hippocampus was higher in the posterior of the brain and less affected anteriorly (Supplementary Fig. 6c–f). Between individuals, the dentate gyrus was more variably affected, with the nuclei sometimes being completely pyknotic, completely unaffected, or intermediately affected (Fig. 1n, o and Supplementary Fig. 6g–j). In addition, we consistently observed pyknotic neuronal nuclei in the cerebral cortex, amygdala, thalamus and motor cortex, though the effect in the amygdala and thalamus was less severe than the hippocampus or cortex. (Fig. 1h, i, p, q and Supplementary

Fig. 6k–r). Within the cerebral cortex, most of the pyknotic nuclei were typically found in layers II/III, IV, and VI (Supplementary Fig. 6k, l). It is possible that the neurodegeneration in the motor cortex contributes to the observed paralysis phenotype. However, at the moment it is not possible to determine definitively if this is the case. Finally, in the cerebral cortex of $Lsd1^{CAGG}$ mice, and to a lesser extent in the hippocampus, we observed a strong reactive gliosis response (Fig. 1t–w), an effect previously associated with neuronal distress[27].

To confirm that the pyknotic nuclei in the hippocampus and cortex of $Lsd1^{CAGG}$ mice have undergone cell death, we performed TUNEL. Nearly every pyknotic nucleus exhibited positive TUNEL staining, indicating that they were undergoing or had undergone cell death (Fig. 1x–aa). Also, the neuronal cell death was observed at the terminal phenotype regardless of the age of the mice when $Lsd1$ was inducibly deleted. These data indicate that LSD1 is continuously required for the survival of hippocampal and cortex neurons.

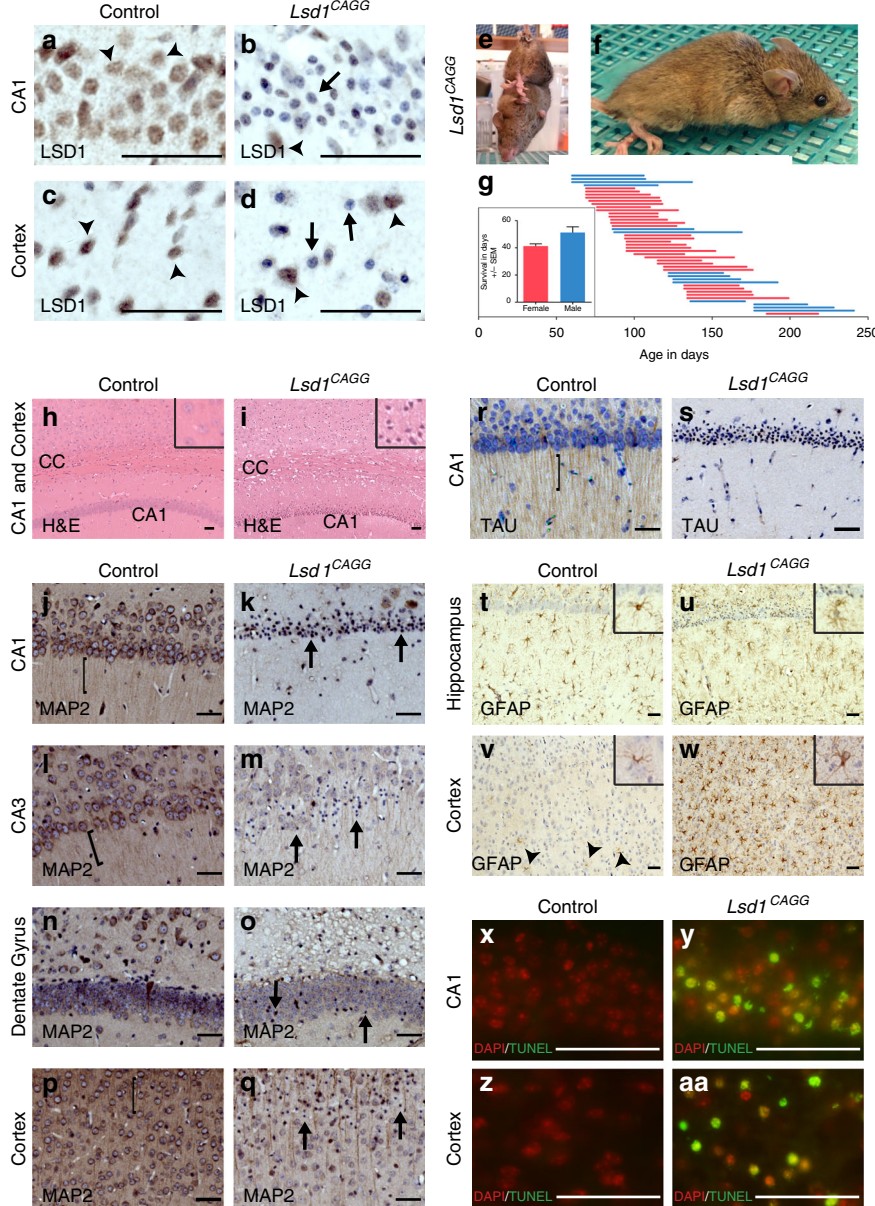

**Fig. 1** Neurodegeneration in $Lsd1^{CAGG}$ mice. **a–d** LSD1 immunohistochemistry (IHC) of control **a**, **c** and $Lsd1^{CAGG}$ **b**, **d** CA1 **a**, **b** and cortex **c**, **d**. Arrowheads highlight non-pyknotic LSD1 immunoreactive nuclei. Arrows highlight pyknotic LSD1 negative nuclei. **e**, **f** Representative images of $Lsd1^{CAGG}$ mice with the terminal motor defect including hindlimb clasping **e** and failure to maintain posture **f**. **g** The age of each individual male (blue) or female (red) mouse at the final tamoxifen injection (start of each line) to inducibly delete $Lsd1$, and the number of days (length of the line) until the terminal motor defect is reached. Inset shows survival in days for each sex. Data are shown as mean survival in days ± s.e.m., $n = 45$ animals **h**, **i** H&E staining of tamoxifen injected $Cre$ minus control (control) **h** and $Lsd1^{CAGG}$ **i** CA1 and cortex. Insets are magnified views of non-pyknotic **h** and pyknotic **i** nuclei. CC denotes corpus callosum. **j–q** MAP2 IHC of control **j**, **l**, **n**, **p** and $Lsd1^{CAGG}$ **k**, **m**, **o**, **q** CA1 **j**, **k**, CA3 **l**, **m**, dentate gyrus **n**, **o** and cortex **p**, **q**. Brackets highlight dendrites and arrows highlight pyknotic nuclei. **r**, **s** Tau IHC of control **r** and $Lsd1^{CAGG}$ **s** CA1. Bracket highlights axons. **t–w** GFAP IHC of control **t**, **v** and $Lsd1^{CAGG}$ **u**, **w** hippocampus **t**, **u** and cortex **v**, **w**. Arrowheads highlight sparse astrocytes in control cortex. Insets show magnified view of representative astrocytes. **x–aa** Merge of DAPI (red) and TUNEL (green) in control **x**, **z** and $Lsd1^{CAGG}$ **y**, **aa** CA1 **x**, **y** and cortex **z**, **aa**. All IHC **j–w** is counterstained with hematoxylin. All $Lsd1^{CAGG}$ images are taken at the terminal phenotype. Scale bar = 50 µm

Immunohistochemistry verified that LSD1 protein is lost in the degenerating neurons of $Lsd1^{CAGG}$ mice. Specifically, LSD1 was undetectable in most cortical nuclei and nearly all hippocampal nuclei, including all of the pyknotic nuclei in both regions (Fig. 1a–d). In contrast, LSD1 persisted in the remaining normal uncondensed nuclei within these brain regions (Fig. 1b, d, Supplementary Fig. 7a–d). The reciprocal relationship between LSD1 protein and pyknotic nuclei indicates that the neuronal cell death is likely due to the cell autonomous loss of LSD1. To confirm that hippocampal neurodegeneration is cell autonomous, we also induced deletion of $Lsd1$ in $Lsd1^{CAGG}$ mice using a single low dose tamoxifen injection. In contrast to the widespread LSD1 protein loss that we observe in the hippocampus with multiple higher dose tamoxifen injections (Supplementary Fig. 7a–d), 10 weeks after tamoxifen injection the low dose injection resulted in the loss of LSD1 protein in only a small number of neurons within the hippocampus (Supplementary Fig. 7e–f). Nevertheless, the few neuronal nuclei that lack LSD1 still become pyknotic, indicating that they have undergone neurodegeneration (Supplementary Fig. 7e–f). These results suggest that within the hippocampus, the neuronal cell death is cell autonomous.

Despite the severe neurodegeneration of the hippocampus and cortex in $Lsd1^{CAGG}$ mice, the cerebellum appeared normal. This can be seen, for example, by the absence of pyknotic nuclei and the normal distribution of the dendrite marker MAP2 (Supplementary Fig. 8a–d). To determine whether the lack of neuronal cell death in this region could be due to the failure of $Lsd1$ deletion there, we performed quantitative PCR to assess the extent of remaining undeleted $Lsd1$ in different brain regions. This analysis demonstrated high levels of deletion in the hippocampus and to a lesser extent in the cerebral cortex. However, there was very little $Lsd1$ deletion in the cerebellum

(Supplementary Fig. 8g). Overall, the extent of deletion matches the level of remaining LSD1 protein in each brain region at the terminal stage, with very little LSD1 in the hippocampus, low levels of LSD1 in the cortex, and higher levels of LSD1 in the cerebellum (Fig. 1a–d and Supplementary Fig. 8e, f). This distribution suggests that the brain region specificity of the neurodegeneration in $Lsd1^{CAGG}$ mice may be due to the specificity of $Lsd1$ deletion. Notably, though $Lsd1$ deletion in the hippocampus occurred within the first 24 h after tamoxifen injection (Supplementary Fig. 8g), the loss of LSD1 protein in the hippocampus occurred much later. For example, in the hippocampi of mice just beginning to display hindlimb weakness (~ 1 week before the terminal phenotype) we observed some remaining LSD1 immunoreactivity and far fewer pyknotic nuclei (Supplementary Fig. 8h, i). This indicates that there is slow RNA or protein turnover in hippocampal neurons, a finding that is consistent with the continuous requirement for LSD1 in these cells.

Many previous mouse models of neurodegeneration display moderate levels of neuronal loss over an extended period of time (many months)[28, 29], so the extent of neuronal cell death that we observed in $Lsd1^{CAGG}$ mice within 9 weeks was striking. Therefore, we considered the possibility that LSD1 is generally required for cell viability. If this were the case, deletion of $Lsd1$ throughout the mouse would be expected to result in a similar disruption in other organs and cell types. To address this possibility, we examined the liver and kidneys of terminal $Lsd1^{CAGG}$ mice using dual IF. Hepatocytes and nephron epithelial cells lacking LSD1 appeared morphologically normal (Supplementary Fig. 9a–l). Additionally, Purkinje neurons lacking LSD1 in the cerebellum did not display any morphological signs of cell death despite the absence of LSD1 (Supplementary Fig. 8e, f).

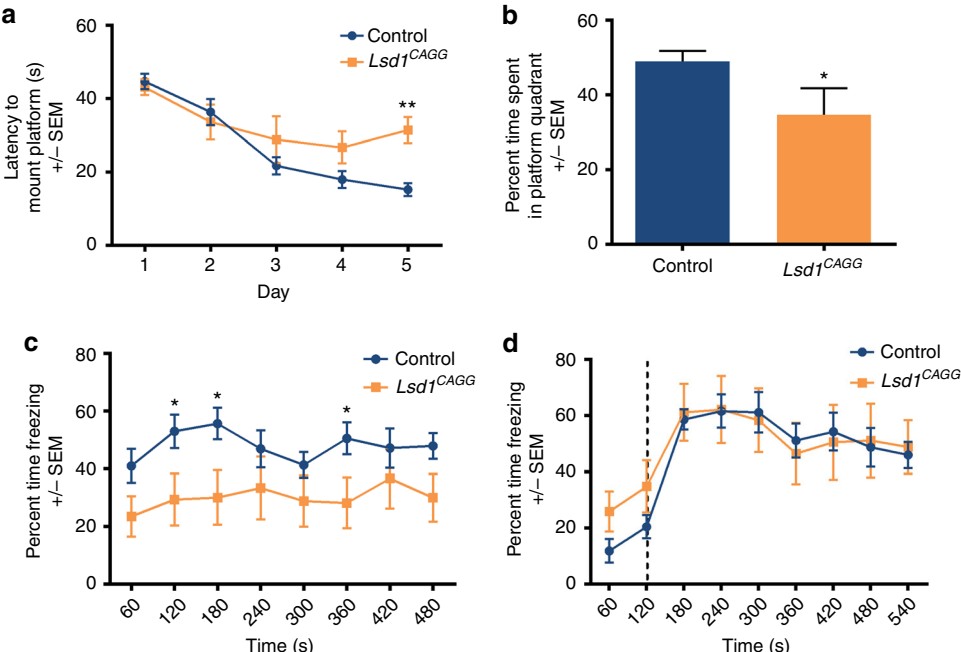

**Fig. 2** Loss of LSD1 results in learning and memory deficits. **a** Latency to mount platform (in seconds) in the Morris water maze across the 5 day training period of control (blue, $n = 15$) and $Lsd1^{CAGG}$ (orange, $n = 12$) mice. Data are shown as mean ± s.e.m. **$P < 0.01$ on day 5 compared by repeated measures two-way ANOVA with post hoc Sidak's multiple comparisons test. **b** Percent time spent swimming in platform quadrant during probe (day 6) after 5 days of water maze training for control (blue $n = 15$) and $Lsd1^{CAGG}$ mice (orange, $n = 11$) mice. Data are shown as mean ± s.e.m. *$P < 0.05$ by unpaired $t$-test. **c** Percent time spent freezing during contextual fear response after fear conditioning of control (blue, $n = 12$) and $Lsd1^{CAGG}$ (orange, $n = 8$) mice. Data are shown as mean ± s.e.m. *$P < 0.05$ by unpaired $t$-test at individual timepoints. $P = 0.052$ for difference between genotypes by repeated measures two-way ANOVA. **d** Percent time spent freezing during cued fear response after fear conditioning of control (blue, $n = 12$) and $Lsd1^{CAGG}$ (orange, $n = 8$) mice. Data are shown as mean ± s.e.m. Dashed line represents sound of tone

Taken together, these data suggest that LSD1 is not required for general cell viability. This conclusion is consistent with what has been reported in the literature elsewhere[4, 6, 9–11, 16, 30]. Thus, the continuous requirement for LSD1 to prevent neuronal cell death in the hippocampus and cortex appears to be specific to these neurons.

**Loss of LSD1 results in learning and memory defects.** To determine whether LSD1-dependent neurodegeneration leads to learning and memory deficits, we assessed female *Lsd1^{CAGG}* mice in the Morris water maze and fear conditioning assays, 28 days after tamoxifen injection (prior to the onset of motor defects). Compared to littermate controls, *Lsd1^{CAGG}* mice had significant defects in the latency to mount the platform in the water maze assay on day 5 (Fig. 2a). This is despite the fact that *Lsd1^{CAGG}* mice swam at speeds not significantly different than their littermate controls (Supplementary Fig. 10a, Supplementary Movie 2). Also, on day 5, there is an increase in overall distance traveled as *Lsd1^{CAGG}* mice swim randomly rather than locating the platform (Supplementary Fig. 10b). Together these results suggest that the impaired performance of *Lsd1^{CAGG}* mice in the water maze is not due to motor deficits. On day 6, when the platform was removed, controls spent nearly half of their time swimming in the platform quadrant, while *Lsd1^{CAGG}* mice spent approximately equal time swimming in each of the four quadrants (Fig. 2b). These data suggest that *Lsd1^{CAGG}* mice have reduced spatial learning and reference memory capacity. *Lsd1^{CAGG}* mice were also impaired in contextual fear conditioning, spending less time freezing ($30.0 \pm 8.3\%$ average with s.e.m.) compared to controls ($47.9 \pm 4.5\%$ average with s.e.m.) (Fig. 2c). The contextual fear conditioning was reduced in *Lsd1^{CAGG}* mice at all points, and this

reduction was statistically significant at 120, 180, and 360 s (Fig. 2c). However, *Lsd1^{CAGG}* mice froze normally in response to a conditioned tone during cued fear conditioning (Fig. 2d). These data suggest that *Lsd1^{CAGG}* mice have defects in contextual, but not cued, learning and memory. This specificity is consistent with the observed pattern of neuronal cell death in the brains of these mice. Notably, though we do not detect any evidence of visual impairment, it is possible that a slight defect in visual impairment also contributes to the deficit observed in the water maze and contextual fear conditioning assays.

**LSD1 inhibits re-activation of stem cell transcription.** Previous work has implicated the LSD1-containing CoREST complex in repressing neuronal genes in non-neuronal cell types[17, 18]. This raised the possibility that LSD1 may be functioning similarly in terminally differentiated hippocampal neurons to block the expression of genes associated with alternative cell fates. To test this possibility, we examined hippocampal gene expression changes in terminal *Lsd1^{CAGG}* mice by RNA-seq. At this terminal stage, there was no difference in the number of pyknotic nuclei in *Lsd1^{CAGG}* mutants vs. the number of normal nuclei in unaffected controls, indicating that neurons in *Lsd1^{CAGG}* were actively undergoing neuronal cell death, but not yet cleared (Supplementary Fig. 11a). Comparison of global gene expression by unsupervised hierarchical clustering and principle components analysis in two *Lsd1^{CAGG}* mutants and two tamoxifen-injected *Cre* minus littermate controls, showed that the expression states were similar between biological replicates, but different between *Lsd1^{CAGG}* mutants and controls (Supplementary Fig. 11b, c and Supplementary Data 1). Also, analysis of differentially expressed genes between *Lsd1^{CAGG}* mutant and control hippocampi

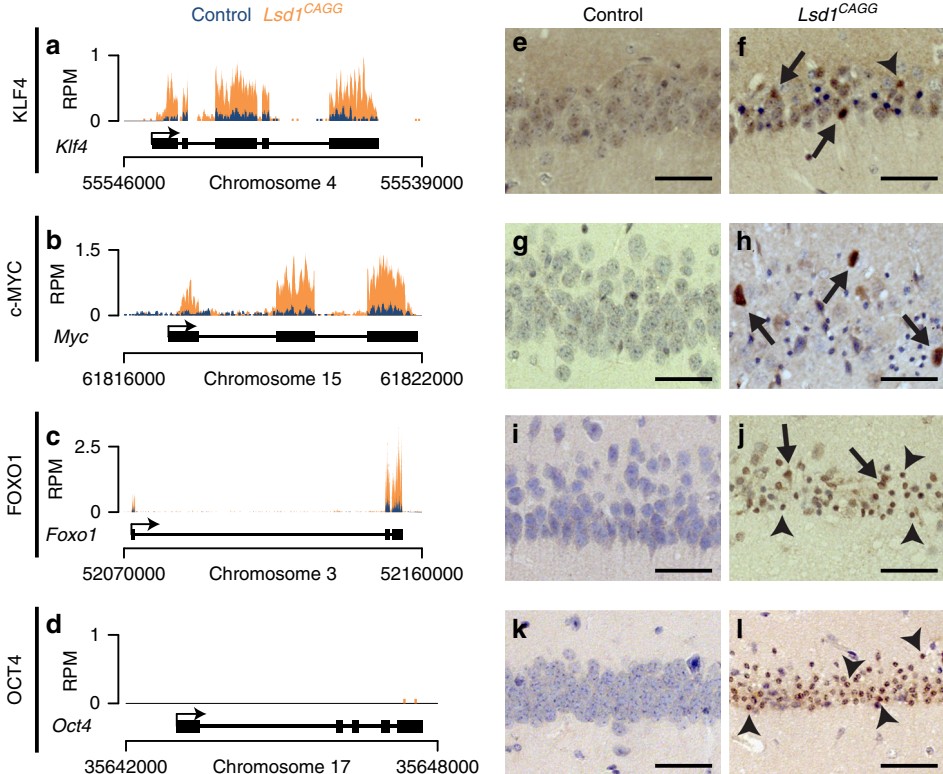

**Fig. 3** Ectopic activation of stem cell genes in *Lsd1^{CAGG}* mice. **a–d** Genome browser style plot of RNA-seq reads per million (RPM) from control (*blue*) and overlaid *Lsd1^{CAGG}* (*orange*) hippocampi showing expression of the genes *Klf4* **a**, *Myc* **b**, *Foxo1* **c**, *Oct4* **d**. **e–l** Immunohistochemistry (IHC) with antibodies to KLF4 **e**, **f**, c-MYC **g**, **h**, FOXO1 **i**, **j**, and OCT4 **k**, **l** in control **e**, **g**, **i**, **k** and *Lsd1^{CAGG}* **f**, **h**, **j**, **l** CA1 neuronal nuclei. *Arrows* denote non-pyknotic nuclei and *arrowheads* denote pyknotic nuclei. All IHC is counterstained with hematoxylin. All *Lsd1^{CAGG}* images are taken at the terminal phenotype. Scale bar = 50 μm

revealed more significantly upregulated (281) than significantly downregulated (124) genes (Supplementary Fig. 11d, e, Supplementary Data 1, FDR < 0.05).

LSD1 has previously been shown to repress the expression of several critical stem cell genes during differentiation in multiple stem cell populations[9, 11, 16]. Therefore, we hypothesized that LSD1 may also be continuously required in terminally differentiated neurons to repress the transcription of stem cell genes to

block the re-initiation of a stem cell fate. To address this possibility, we examined the expression of stem cell genes in our $Lsd1^{CAGG}$ hippocampus RNA-seq data set (Supplementary Data 1). Remarkably, three pluripotency genes (*Klf4, Myc*, and *Foxo1*), two of which are induced pluripotent stem cell (iPSC) factors[31], were among the most significantly upregulated genes in $Lsd1^{CAGG}$ mice (Fig. 3a–c and Supplementary Data 1). Immunohistochemistry (IHC) analysis confirmed that KLF4

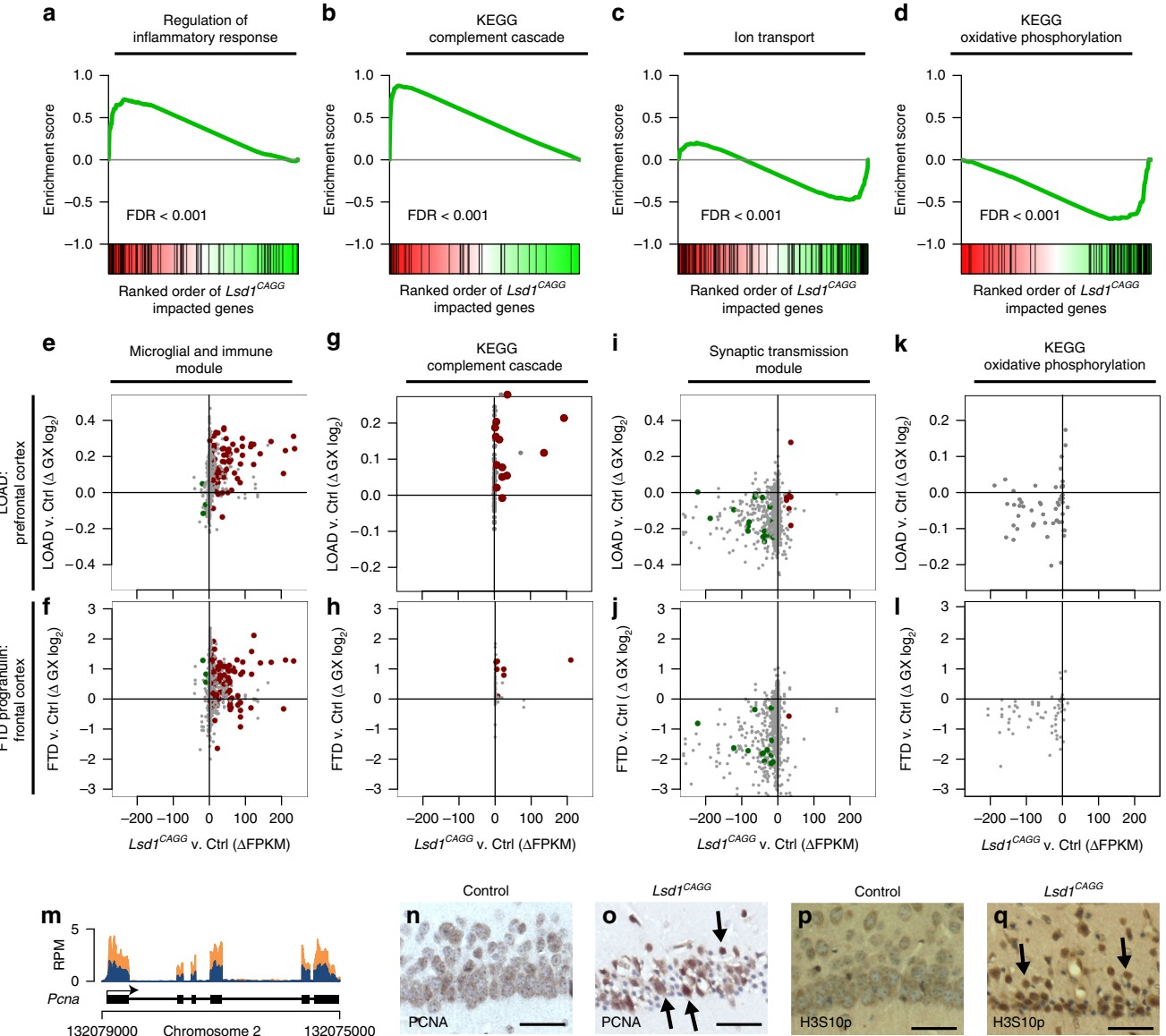

**Fig. 4** Loss of LSD1 induces common neurodegeneration pathways. **a–d** Gene set enrichment plots of neurodegeneration pathways where $Lsd1^{CAGG}$ impacted transcripts (*x*-axis) are sorted by magnitude of upregulation (red) to downregulation (green). The position of each gene from the gene set is represented as a black tick mark (*x*-axis). Enrichment score (*y*-axis) shows where enrichment of genes from the set occurs in the $Lsd1^{CAGG}$ transcriptome. Gene sets shown are regulation of inflammatory response **a**, Kyoto Encyclopedia of Genes and Genomes (KEGG) complement cascade **b**, ion transport **c**, and KEGG oxidative phosphorylation **d**. FDR is shown for each plot. **e–l** Scatter plots showing correlated changes in gene expression of genes from the Microglial and Immune Module[36] **e**, **f**, KEGG complement cascade **g**, **h**, Synaptic Transmission Module[36] **i**, **j** and KEGG oxidative phosphorylation **k**, **l** gene sets between the $Lsd1^{CAGG}$ and control hippocampus (FPKM, *x*-axes) compared with the changes in log₂ gene expression between late onset AD (LOAD) and control prefrontal cortex[36] (**e**, **g**, **i**, **k**; *y*-axis), or compared with the changes between FTD-progranulin and control frontal cortex[37] (**f**, **h**, **j**, **l**; *y*-axis). The most significantly changed genes in the $Lsd1^{CAGG}$ hippocampus (Supplementary Fig. 11d, e) are shown in red (upregulated) and green (downregulated). All other genes with a direct mouse/human orthologue are shown in gray. Genes with correlated expression changes are found in the top right and bottom left quadrants, while genes that do not correlate are found in the other quadrants. **m** Genome browser style plot (as described in Fig. 3a–d) showing *Pcna* expression in $Lsd1^{CAGG}$ hippocampus (orange) compared with control (blue). **n–q** Immunohistochemistry with antibodies to PCNA **n**, **o**, and H3S10p **p**, **q** in control **n**, **p** and $Lsd1^{CAGG}$ **o**, **q** CA1 neuronal nuclei. Arrows denote non-pyknotic nuclei. All IHC is counterstained with hematoxylin. All $Lsd1^{CAGG}$ images are taken at the terminal phenotype. Scale bar = 50 μm

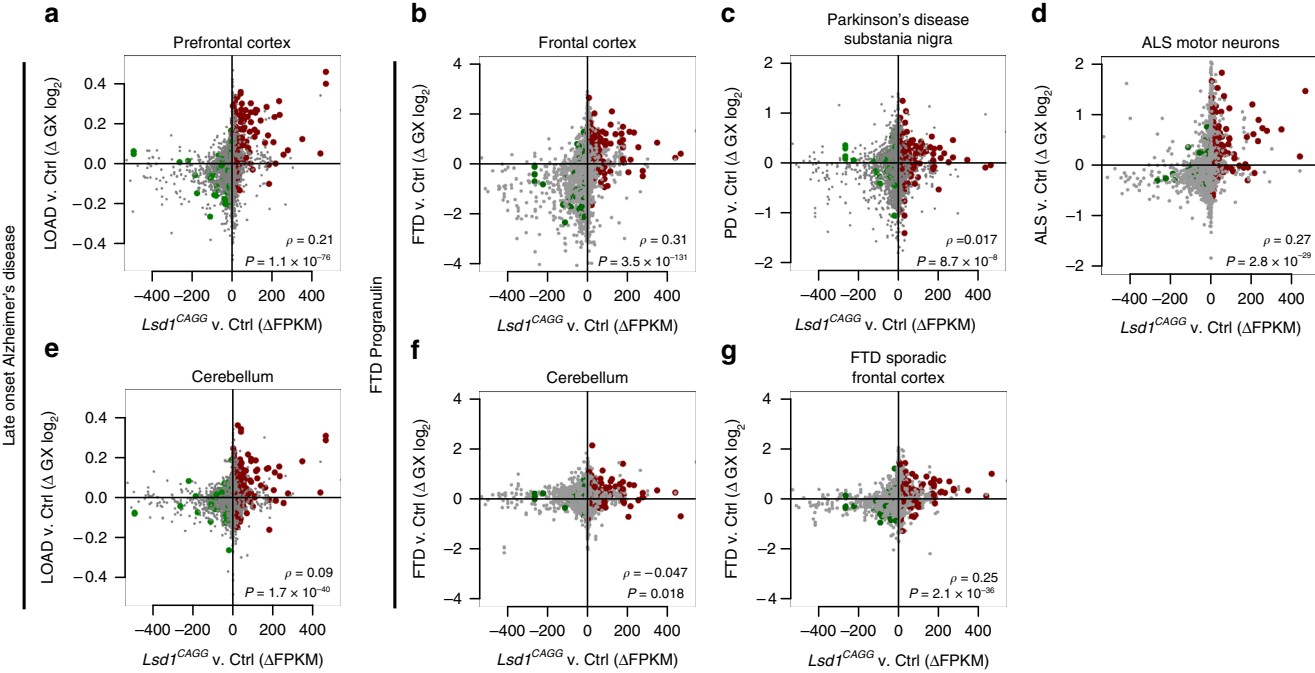

**Fig. 5** Expression changes in $Lsd1^{CAGG}$ mice correlate with those in AD and FTD. **a–f** Scatter plots (as described in Fig. 4e–l) showing genome-wide correlated changes in gene expression between the $Lsd1^{CAGG}$ and control hippocampus (FPKM, x-axes) compared to $log_2$ gene expression changes in late onset AD (LOAD) prefrontal cortex[36] (**a**; y-axis), FTD-progranulin frontal cortex[37] (**b**; y-axis), PD substantia nigra[39] (**c**; y-axis), ALS motor neurons[40] (**d**; y-axis), LOAD cerebellum[36] (**e**; y-axis), FTD-progranulin cerebellum[37] (**f**; y-axis), and sporadic FTD frontal cortex[37] (**g**; y-axis). P-values and ρ Pearson correlation coefficient are given

and FOXO1 proteins were reactivated widely in the degenerating pyknotic neurons, as well as in some of the remaining non-condensed nuclei, but not in controls (Fig. 3e, f, i, j). In contrast, c-MYC was reactivated only in a few nuclei (Fig. 3g, h). Therefore, to confirm that these c-MYC positive cells are neurons, we performed dual IF with the neuronal marker NeuN. This analysis confirmed that c-MYC is reactivated in neuronal nuclei (Supplementary Fig. 12a–h). Interestingly, although we did not observe increased *Oct4* expression in our RNA-seq data set (Fig. 3d), one out of four mice analyzed displayed re-activation of OCT4 protein throughout the pyknotic hippocampal nuclei (Fig. 3k, l). This expression pattern appeared to be specific, as it was not observed in any of the controls or in other brain regions of the affected animal. These results suggest that LSD1 is continuously required to repress the inappropriate expression of stem cell genes in hippocampal neurons.

Among the most highly activated genes in our RNA-seq data set we also noticed the upregulation of the neuronal stem cell genes *Vimentin* and *Nestin* (Supplementary Fig. 12i, j). To determine whether VIMENTIN and NESTIN may also be reactivated in the dying neurons of $Lsd1^{CAGG}$ mice, we performed IHC to detect the expression of these proteins. IHC detected VIMENTIN protein in a subset of hippocampal neurons, though at a higher frequency in $Lsd1^{CAGG}$ mice than controls, while NESTIN protein is found in the reactive glia of the $Lsd1^{CAGG}$ hippocampus and cortex (Supplementary Fig. 12k–p).

**Loss of LSD1 induces common neurodegeneration pathways**. To identify additional pathways associated with the hippocampal neuronal cell death, we also performed gene ontology (GO) and gene set enrichment analysis (GSEA) on our RNA-seq data sets. Among the pathways that are affected by the loss of LSD1, we observed the upregulation of inflammatory response genes

and complement cascade genes, along with the downregulation of oxidative phosphorylation genes and genes involved in neurotransmission (ion transport) (Supplementary Fig. 11f, g). All four of these pathways have been previously linked to neurodegeneration. For example, several studies have implicated the inflammatory response pathway in neurodegeneration. Activation of the inflammatory response pathway could contribute to neurodegeneration via macrophage mediated phagocytosis[32]. There is also evidence linking the complement cascade pathway to neurodegeneration. Activation of the complement cascade pathway could lead to neuronal cell death through axonal pruning[33]. In addition, impaired neurotransmission could contribute to neuronal cell death through the loss of electrical potential[34]. Finally, a defect in oxidative phosphorylation, with the accompanying mitochondrial dysfunction, could lead to neurodegeneration via the generation of reactive oxygen species[35]. To determine the extent that these four neurodegeneration-associated pathways are misregulated in our $Lsd1^{CAGG}$ hippocampus RNA-seq, we plotted the enrichment of these gene sets in our data set for each of these four pathways. This analysis demonstrated that all four of these common neurodegeneration pathways are highly affected (Fig. 4a–d). Importantly, while each of these pathways has been implicated in neurodegeneration, it is difficult to determine whether these pathways contribute to neuronal cell death, or whether they may simply be a consequence of the neurodegeneration.

$Lsd1^{CAGG}$ **expression changes overlap with AD and FTD cases**. The common neurodegeneration pathways affected by loss of LSD1 are also affected in human neurodegeneration patients. For example, systems biology approaches in human late onset Alzheimer's disease (LOAD) brains have identified a critical microglia and immune transcription network upregulated in AD

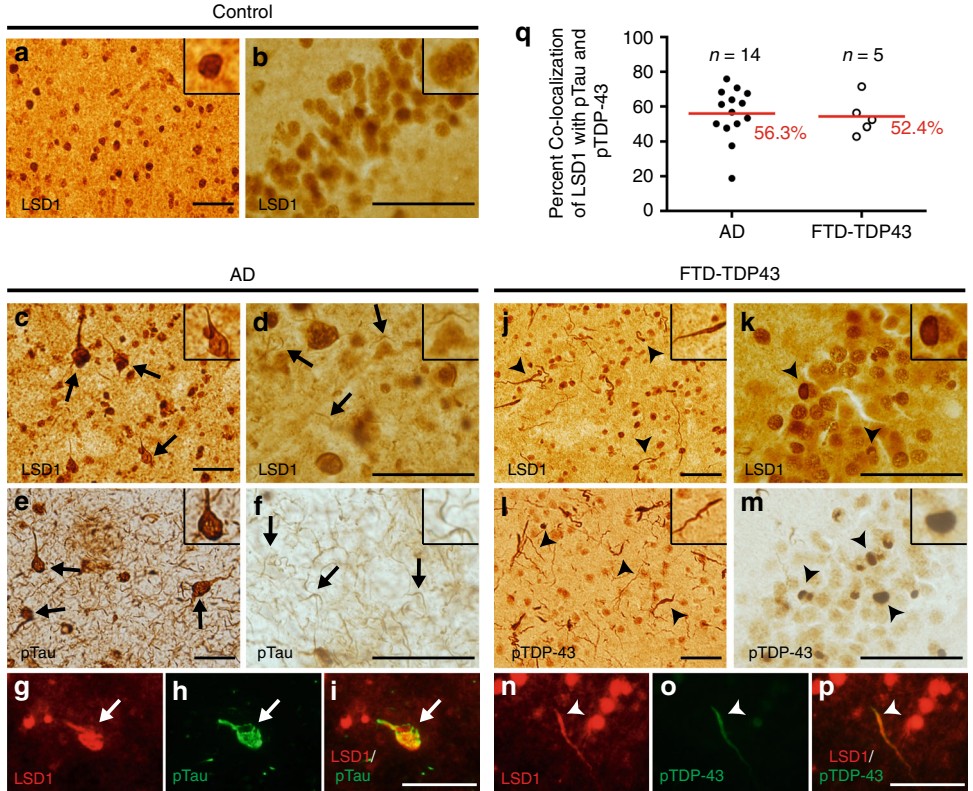

**Fig. 6** LSD1 co-localization with pTau and pTDP-43 aggregates in AD and FTD. **a, b** LSD1 immunohistochemistry (IHC) showing expression of LSD1 in age-matched control frontal cortex **a** and hippocampus **b**. **c, d** Representative IHC images showing LSD1 immunoreactivity localized to cytoplasmic tangle-like aggregates (**c**, arrows) and neurites (**d**, arrows) in AD frontal cortex. **e, f** IHC images showing pTau (AT8 epitope) neurofibrillary tangles (**e**, arrows) and neuropil threads (**f**, arrows) from the same AD frontal cortex as **c**, **d**. **g–i** Representative image of LSD1 (**g**, red), pTau (**h**, green), and merged **i** immunofluorescence (IF) showing co-localization of LSD1 with a pTau neurofibrillary tangle in AD (arrow). **j, k** Representative IHC image showing LSD1 immunoreactivity localized to abnormal deposits in neurites (**j**, arrowheads) and cytoplasmic inclusions (**k**, arrowheads) in FTD-TDP43 frontal cortex **j** and hippocampus **k**. **l, m** IHC images showing pTDP-43 in neurites and cytoplasmic inclusions (**l**, **m**, arrowheads) from the same FTD-TDP43 frontal cortex **l** and hippocampus **m** as **j** and **k**, respectively. **n–p** Representative image of LSD1 (**n**, red), pTDP-43 (**o**, green) and merged **p** IF showing co-localization of LSD1 with pTDP-43 in a neurite in FTD-TDP43 (arrowhead). Insets are magnified views of LSD1 nuclear localization **a**, **b** and representative pathologies **c–f**, **j–m**. Scale bar = 50 μm. **q** The percentage of neurofibrillary tangles (pTau) with LSD1 co-localization in AD (*n* = 14 cases assayed, closed circles), and neurites (pTDP-43) with LSD1 co-localization in FTD-TDP43 (*n* = 5 cases assayed, open circles), with the average percentage shown (red bar)

cases[36]. Interestingly, we noticed that many genes in the LOAD microglia and immune gene signature, including the critical receptor *Tyrobp*, are highly enriched in the *Lsd1*[CAGG] hippocampus (Supplementary Data 1). Also, many of these microglia and immune genes are among the 281 most significantly upregulated genes in our RNA-seq data set (Supplementary Data 1). Therefore, to determine if the LOAD microglia and immune response module is similarly misregulated in our mice, we compared the expression changes in the *Lsd1*[CAGG] hippocampus to previously published expression changes at orthologous loci in LOAD cases[36]. This analysis demonstrated that loss of LSD1 in the mouse hippocampus leads to microglia and immune response gene expression changes that are highly similar to those that occur in the prefrontal cortex of LOAD cases. The microglia and immune expression changes in the *Lsd1*[CAGG] hippocampus also highly overlap with those that occur in the frontal cortex of FTD cases with progranulin mutations (FTD-progranulin)[37] (Fig. 4e, f).

Surprisingly, a similar correlation with AD and FTD cases is also found with the other neurodegeneration pathways that are misregulated in our RNA-seq data set. For example, in the Kyoto Encyclopedia of Genes and Genomes (KEGG) complement cascade genes, expression changes in the *Lsd1*[CAGG] hippocampus highly overlap with the upregulation that occurs in the prefrontal cortex of AD and FTD cases (Fig. 4g, h). A correlation is observed in pathways that are downregulated in the *Lsd1*[CAGG] hippocampus as well. For example, we find a large overlap with expression changes in the neurotransmission genes (Synaptic Transmission Module) that were also identified using systems biology approaches in LOAD cases (Fig. 4k, l)[36]. Similarly, we observe a high correlation with the transcriptional changes in oxidative phosphorylation genes (Fig. 4i, j).

Finally, among the top upregulated genes in the *Lsd1*[CAGG] hippocampus we noticed the cell cycle gene *PCNA* (Fig. 4m, and Supplementary Data 1). Evidence for the potential re-initiation of the cell cycle has been found in AD cases[38]. Therefore, to determine if PCNA, and other cell cycle markers, are being reactivated in degenerating *Lsd1*[CAGG] neurons, we performed IHC analysis. This analysis confirmed the re-activation of PCNA protein, along with that of another cell cycle marker, H3S10p, specifically in the remaining non-pyknotic hippocampal nuclei (Fig. 4n–q). Intriguingly, the observation that c-MYC, PCNA and H3S10p were reactivated predominantly in the remaining uncondensed nuclei of the *Lsd1*[CAGG] hippocampus raises the possibility that these neurons may be attempting to re-initiate the cell cycle prior to neuronal cell death.

The high degree of overlap within multiple neurodegeneration pathways between *Lsd1*[CAGG] mice and human dementia cases

was unexpected. Thus, we considered the possibility that the expression changes in our mice might overlap more broadly with AD and FTD cases. To address this possibility, we next compared the expression changes in the $Lsd1^{CAGG}$ hippocampus with the expression changes in AD and FTD cases genome-wide. Remarkably, we found that the genome-wide expression changes in the prefrontal cortex of LOAD cases highly correlate with the expression changes in the hippocampus of $Lsd1^{CAGG}$ mice (Fig. 5a). Likewise, the correlation was highly significant when compared to the frontal cortex of FTD-progranulin (Fig. 5b).

The genome-wide correlation in expression changes with AD and FTD cases could indicate the possible involvement of LSD1 in these diseases. However, it is also possible that the overlap is being primarily driven by the consequences of neuronal cell death. To address this second possibility, we compared the expression changes in the $Lsd1^{CAGG}$ hippocampus with other neurodegenerative diseases that have similar levels of neuronal cell death. If the genome-wide correlation is being driven by a common underlying mechanism, rather than neuronal cell death, we would expect the correlation to be less significant in these comparisons. Importantly, we observe relatively little overlap with the expression changes in the substantia nigra of Parkinson's disease (PD), a region with extensive neuronal cell death (Fig. 5c)[39]. We also see relatively little overlap with the expression changes that occur in the motor neurons of amyotrophic lateral sclerosis (ALS) cases (Fig. 5d)[40]. Furthermore, compared with the high degree of correlation that we observe in FTD-progranulin cases, we find a dramatic reduction in the correlation when compared to sporadic FTD cases, despite the fact that these sporadic FTD cases have levels of neuronal cell death that are the same as FTD-progranulin cases (Fig. 5g). The large decrease in gene expression overlap, that we observe in PD, ALS, and sporadic FTD cases, suggests that the genome-wide overlap in expression with AD and FTD cases, is not simply due to neuronal cell death. Finally, we also compared $Lsd1^{CAGG}$ hippocampus expression changes to changes in the cerebellum of AD and FTD cases. Compared with the prefrontal cortex, the cerebellum is relatively unaffected in AD and FTD cases. In both AD and FTD, we find that expression changes in the cerebellum overlap much less than the prefrontal cortex (Fig. 5e, f). This discrepancy indicates that within AD and FTD cases, the overlap in expression may be driven by the neurodegeneration, rather than brain region.

**LSD1 is mislocalized in human dementias**. The RNA-seq data suggest that deletion of $Lsd1$ alone is sufficient to recapitulate transcriptional changes observed in the affected brain regions of AD and FTD-progranulin cases, including many of the individual gene categories that have previously been implicated in the etiology of these dementias. These data potentially implicate the loss of LSD1 function in these human dementias. As a result, we wondered whether LSD1 might be affected in AD and FTD patients. AD is characterized by protein aggregates of amyloid β (Aβ) and Tau, while FTD is associated with aggregates of either Tau or Tar DNA binding protein 43 (TDP-43)[41–43]. These pathological aggregates are thought to lead to downstream pathways of neurodegeneration, but it remains unclear mechanistically how these aggregates are linked to neuronal cell death.

To determine whether LSD1 may be affected in AD and FTD patients, we examined the localization of LSD1 in post-mortem AD, FTD with TDP-43 inclusions (FTD-TDP43), and age-matched control cases. We also examined the localization of LSD1 in Parkinson's disease (PD) cases, as a disease control with pathological protein aggregates. Similar to the expression in mice, LSD1 immunoreactivity was found in neuronal nuclei throughout

the frontal cortex and hippocampus of age-matched control cases (Fig. 6a, b). In contrast, in all 14 AD cases analyzed, LSD1 was found both in neuronal nuclei as well as inappropriately associated with cytoplasmic tangle-like aggregates and neurites, (Fig. 6c, d). This pattern is highly reminiscent of the neurofibrillary tangles and neuropil threads marked by pTau in the same AD cases (Fig. 6c–f). In addition, in all 14 FTD-TDP43 cases analyzed, LSD1 was abnormally associated with neurites in the frontal cortex, and cytoplasmic inclusions in the hippocampus (Fig. 6j, k). This pattern is highly similar to the pTDP-43 aggregation observed in FTD-TDP43 cases (Fig. 6j–m). To confirm the co-localization of LSD1 with pTau and pTDP-43 we performed dual IF. This analysis demonstrated that LSD1 co-localizes with pTau in 56.3% of neurofibrillary tangles in AD ($n = 14$ patients), and with pTDP-43 in 52.4% of neurites in FTD-TDP43 ($n = 5$ patients) (Fig. 6g–i, n–q). Within AD cases the extent of co-localization ranges from 19 to 76%, while in FTD-TDP43 cases, the co-localization ranges from 43 to 71%. The finding that LSD1 is localized to pathological aggregates raises the possibility that it could be increasingly sequestered in the cytoplasm. This could result in less LSD1 being available to function in the nucleus of affected neurons in AD and FTD cases.

To confirm the specificity of the LSD1 localization, we performed several controls. Preincubation of the LSD1 Antibody (Ab) with its target LSD1 peptide completely abrogated the immunoreactivity (Supplementary Fig. 13a, b). We also did not observe the localization of LSD1 to the amyloid β core of senile plaques in the same AD cases where we observed co-localization with pTau (Supplementary Fig. 13c, d). Nor do we observe LSD1 localized to any Lewy body-like structures (aggregates of α-synuclein), or any other abnormal localization of LSD1, in the substantia nigra of PD cases (Supplementary Fig. 13e–h). These results suggest the mislocalization of LSD1 to neurofibrillary tangles in AD, and pTDP-43 inclusions in FTD cases, is specific. Notably, the co-localization of proteins with these pathological aggregates is exceedingly rare. For example, though many proteins have been recently described as enriched in the insoluble fraction of AD brains, only one was confirmed to be co-localized with neurofibrillary tangles[44].

**$Lsd1^{CAGG}$ mice do not have protein aggregates**. Since LSD1 associates with pathological aggregates in AD and FTD-TDP43 cases, we considered the possibility that the neuronal cell death that we observe in the $Lsd1^{CAGG}$ mice could be due to the induction of pathological aggregates in the mice. To test this possibility, we performed IHC on the brains of terminal $Lsd1^{CAGG}$ mice using antibodies to Aβ, pTau, and pTDP-43, along with Gallyas (silver, nonspecific aggregates) staining (Supplementary Fig. 14a–h). We find no evidence of any pathological protein aggregates or tangles associated with the degenerating neurons or otherwise. This suggests that if loss of LSD1 is involved in AD and/or FTD, it is likely downstream of pathological aggregation. This finding is consistent with the mislocalization of LSD1 to pathological aggregates in the human cases (Fig. 6).

**Increased stem cell gene expression in AD and FTD patients**. The loss of LSD1 in mice is associated with the surprising re-activation of stem cell transcription in hippocampal neurons. If LSD1 is affected in AD and/or FTD, these diseases could be associated with a similar increase in stem cell gene expression. To test this possibility, we re-examined the expression of stem cell genes in previously published microarray experiments from LOAD and FTD-progranulin post-mortem cases[36, 37]. This analysis revealed a significant increase in the expression of $Klf4$, $Myc$,

*Oct4*, *Foxo1*, and *Vimentin* in LOAD cases compared to controls (Supplementary Fig. 15a, c, e, g, k), while *PCNA* expression was unchanged (Supplementary Fig. 15i). In FTD-progranulin cases there was also a significant increase in the expression of *Klf4* and *Foxo1*, as well as a trend toward the increased expression of *Myc*, *Oct4*, *PCNA*, and *Vimentin* (Supplementary Fig. 15b, d, f, h, j, l). These data are consistent with the possibility that LSD1 function could be compromised in AD and FTD patients.

## Discussion

Despite its well-known role throughout development, LSD1 protein can also be found in terminally differentiated cells throughout the brain. To determine whether there is an ongoing role for LSD1 in these terminally differentiated cells, we conditionally deleted *Lsd1* in adult mice. Surprisingly, within the brain at the terminal time point, the inducible loss of LSD1 in *Lsd1*$^{CAGG}$ mice results in loss of LSD1 protein only in neurons. This indicates that neurons may be more vulnerable to LSD1 protein or RNA turnover, a specificity that mirrors what occurs in AD and FTD cases.

Within the brain, the selective vulnerability of neurons in *Lsd1*$^{CAGG}$ mice enables us to specifically interrogate the function of LSD1 in these cells. Loss of LSD1 in *Lsd1*$^{CAGG}$ mice results in widespread hippocampus and cortex neuronal cell death. This demonstrates that loss of LSD1 in hippocampus and cortex neurons is sufficient to induce neuronal cell death. This conclusion is consistent with our high/low tamoxifen mosaic experiments, which indicate that LSD1 acts cell autonomously in hippocampal neurons. Thus, we propose that LSD1 functions continuously in hippocampal and cortex neurons to prevent neurodegeneration.

To further investigate the neuronal cell death in the hippocampus, we examined gene expression changes genome-wide. Previous analyses of human neurodegeneration cases and experimental models have implicated common pathways leading to neuronal cell death. These include; activation of genes in the microglia and immune pathways, a defect in oxidative phosphorylation, loss of synaptic transmission, and failure to maintain cell cycle arrest. Remarkably, the loss of LSD1 affects all of these common neurodegenerative pathways. Therefore, it is possible that the loss of LSD1 creates a perfect storm where multiple neurodegenerative pathways are affected simultaneously, with one or more of these pathways leading to the observed neuronal cell death.

The prevailing view in developmental biology is that cells are irreversibly committed to their differentiated cell fate. Indeed, the very word "fate" promotes the idea that a differentiated cell has reached its final destiny. However, there may be a requirement for differentiated cells to actively maintain their differentiated status. The LSD1-containing CoREST complex has been previously implicated in repressing neuronal genes in non-neuronal cells[17, 18]. Based on this, we considered the possibility that LSD1 could be similarly required to maintain terminally differentiated hippocampus and cortex neurons by repressing gene transcription associated with alternative cell fates. In the degenerating neurons of *Lsd1*$^{CAGG}$ mice, we detect the re-activation of stem cell transcription factors, such as KLF4, OCT4, c-MYC and FOXO1. This demonstrates that LSD1 is continuously required in terminally differentiated neurons to block the re-activation of these factors. Also, we detect a widespread decrease in the expression of neuronal pathways. This suggests that LSD1 is also required, directly or indirectly, to maintain the expression of these genes. Therefore, we propose that LSD1 is a key component of an epigenetic maintenance program that reinforces the differentiated state of hippocampal neurons by continuously restraining the re-activation of factors associated with alternative cell fates.

At this moment, we cannot definitively determine why the loss of LSD1 results in a severe motor defect. Nevertheless, *Lsd1*$^{CAGG}$ mice develop a motor defect that is similar to a tauopathy mouse model[28]. For example the P301S mice, which overexpress an aggregation prone form of human Tau, have a motor defect that is reminiscent of *Lsd1*$^{CAGG}$ mice[28]. The concordance of phenotypes between P301S mice and *Lsd1*$^{CAGG}$ mice is consistent with Tau and LSD1 acting in a common pathway. Also consistent with this possibility, we find that that LSD1 inappropriately mislocalizes to cytoplasmic aggregates of pTau in AD, and global gene expression changes in the degenerating *Lsd1*$^{CAGG}$ hippocampus correlate with changes in AD and FTD-progranulin cases. Finally, the re-examination of stem cell genes that are specifically affected by the loss of LSD1 in the mouse hippocampus demonstrates that these genes are also increased in AD and FTD cases. Together these data indicate a potential link between the loss of LSD1 and these human dementia cases. This could occur through the following potential model: as neurons age, the accumulation of protein aggregates sequesters LSD1 in the cytoplasm, and interferes with the continuous requirement for LSD1. Normally, LSD1 maintains terminally differentiated neurons, and prevents the activation of common neurodegenerative pathways, by continuously repressing the transcription of inappropriate genes. As a result, the inhibition of LSD1 by the pathological aggregates in the aging neurons of AD and FTD brains creates a situation where neurons are subject to an onslaught of detrimental processes. This results in neuronal cell death and dementia.

## Methods

**Tamoxifen injections**. All mouse work, including surgical procedures, were approved by and conducted in accordance with the Emory University Institutional Animal Care and Use Committee. Mice were intraperitoneally injected with 75.0 mg tamoxifen per kilogram of body mass dissolved in corn oil once a day on days 1, 2, 4, 5, and 7 of a 7 day period. This injection protocol was used for all assays, except the single low dose tamoxifen injection, which was performed with 1 mg tamoxifen per 40 g of body mass.

**Mouse tissue fixation**. Mice were given a lethal dose of isoflurane via inhalation, then transcardially perfused with ice-cold 4.0% paraformaldehyde in 0.1 M phosphate buffer. Brain, spinal cord and muscle tissues were dissected and post fixed in cold paraformaldehyde solution for 2 h. Tissues were then either cryoprotected by sinking in 30% sucrose and frozen embedded in O.C.T. Compound (Tissue Tek), or serially dehydrated and embedded in paraffin.

**Histology**. For hematoxylin and eosin, and thionin staining was performed according standard procedures (1% thionin, pH4.0). For Gallyas staining, sections were dewaxed then treated with 5% periodic acid for 5 min, followed by washing and 1 min alkaline silver treatment. Sections were then developed for 30 min, followed by 0.5% acetic acid and water rinses. Finally, sections were treated with Schiff's reagent for 30 min, washed, counterstained with hematoxylin and coverslipped.

**Mouse immunofluorescence**. Frozen mouse brain tissue was sectioned at 12 μm and washed with TBS, then treated with 0.8% sodium borohydride for 10 min to reduce background. Antigen retrieval was performed by microwaving at 10% power 3X for 5 min in 0.01 M sodium citrate. Slides were then cooled and washed with TBS, then permeabilized in 0.5% Triton X-100 for 20 min, followed by blocking in 10% goat serum for 1 h. Primary Abs (Supplementary Table 1) were incubated overnight at 4 °C. Slides were then washed and incubated in secondary Abs (Goat α Mouse Alexa Fluor 488, 1:500, Invitrogen A11001 and Goat α Rabbit Alexa Fluor 594, 1:500, Invitrogen A11012) for 1 h at room temperature followed by washes with TBS and DAPI, then coverslipped.

**Mouse immunohistochemistry**. Paraffin embedded tissue was dewaxed with xylenes and serial ethanol dilutions then treated with 3% hydrogen peroxide at 40 °C for 5 min to quench endogenous peroxidase activity, blocked in 2% serum at 40 °C for 15 min, and incubated with primary Ab (Supplementary Table 1) overnight at 4 °C. Slides were washed then incubated with biotinylated secondary Ab (Biotinylated Goat α Rabbit, 1:200, Vector Labs BA-1000 and Biotinylated Goat α

Mouse, 1:200, Vector Labs BA-9200) at 37 °C for 30 min. Signal amplification was then carried out by incubating at 37 °C for 1 h with Vector Labs Elite ABC reagent (PK-6200). Slides were then developed with DAB for 2–5 min, counterstained with hematoxylin and coverslipped.

**Human immunohistochemistry.** The post-mortem human samples were provided by the Alzheimer's Disease Research Center (ADRC) Brain Bank at Emory University and were obtained with signed informed consent in accordance with institutional guidelines.

Frozen free floating sections of 20–50 µm thickness were washed of cryoprotectant followed by quenching of endogenous peroxidase activity by incubating in 3% hydrogen peroxide for 15 min. Sections were permeabilized and blocked in 0.1 Triton X-100 and 8% goat serum, then incubated with primary Abs (Supplementary Table 1) overnight at 4 °C followed by incubation with biotinylated secondary Ab (Biotinylated Goat α Rabbit, 1:200, Vector Labs BA-1000 and Biotinylated Goat α Mouse, 1:200, Vector Labs BA-9200) and amplification with Vector Labs Elite ABC reagent. Sections were then treated with DAB (Sigma) for 3–4.5 min then mounted on slides, dried overnight and serially dehydrated and coverslipped with Permount.

**Human immunofluorescence.** For immunofluorescence, sections were prepared as with IHC, except that tissue was incubated with two primary antibodies (Supplementary Table 1) overnight, and with two secondary antibodies, fluorescent goat anti-mouse (1:500, Invitrogen A11001) and biotinylated goat anti-rabbit (1:200, Vector Labs BA-1000). Fluorescent signal amplification was carried out with Vector Labs Elite ABC reagent and developed with PerkinElmer TSA Plus Cyanine 3 System diluted 1:100. Sections were treated with autofluorescence inhibitor (Millipore 2160) after mounting on slides.

**Immunohistochemistry with peptide block.** For peptide block, the LSD1 primary antibody (1:500, Abcam 17721) was preincubated for 24 h at 4 °C with 74-fold molar excess of target peptide (Abcam 17763).

**TUNEL assay.** Frozen embedded brain tissue was sectioned at 12 µm thickness. Slides were permeabilized in 0.1% Triton X-100, 0.1% sodium citrate for 2 min followed by washes in PBS. For antigen retrieval, slides were microwaved for 1 min at 10% power in preboiled 0.1 M sodium citrate then rapidly cooled by adding deionized water. Slides were then washed in PBS and blocked for 30 min at room temperature in 0.1 M Tris, 3.0% BSA, 10% goat serum. Slides were washed twice then incubated with 50 µL of TUNEL labeling solution (Roche In situ Cell Death Detection Kit, Fluorescein) for 1 h at 37 °C. Slides were then washed with PBS and DAPI (0.5 mg/ml) then coverslipped.

**Neuromuscular junctions.** Frozen tibialis anterior tissue was sectioned at 20 µm and blocked with 10% goat serum for 1 h. Slides were then incubated with SV2 primary antibody (1:50, Developmental Studies Hybridoma Bank) in blocking buffer for 72, replacing antibody every 24 h. Slides were then washed and incubated with secondary antibody (1:500, Invitrogen A11001) for 2 h, washed, then incubated with rhodamine conjugated α-bungarotoxin (Life Technologies T-1175) for 1 h, washed and coverslipped.

**Quantification of LSD1 co-localization with pTau and pTDP-43.** Three random fields per section that contained NFTs marked by pTau at 20X and pTDP-43 inclusions at 40X were manually examined. Beginning with the pTau/pTDP-43 fluorescence channel, each aggregate structure was visually inspected. Then, the microscope was switched to the LSD1 fluorescence channel and inspected for LSD1 signal. Structures were scored as positive for LSD1 co-localization if the LSD1 staining pattern was localized to a majority of the aggregate structure. 535 NFTs and 103 pTDP inclusions were scored.

**Motor neuron counting.** Thionin stained spinal cord sections were imaged and cropped to include just the ventral horn. Motor neurons were manually counted by appearance for each image then calculated as a percent of the total number of nuclei, which were counted by converting the image to binary then counting particles using ImageJ.

**CA1 nuclei counting.** Hematoxylin or DAPI stained nuclei of hippocampal sections were scored as normal, pyknotic, or intermediate. Counts were limited to the most dorsal region of the CA1 in the field of a Zeiss Axiophot ocular graticule grid. Three randomly selected sections were counted per animal. Investigator was blind to genotype.

***Lsd1* deletion quantification.** Intact *Lsd1* alleles from hippocampus, cortex, and cerebellum were quantified from phenol extracted genomic DNA using BioRad CFX96 Real-Time System using the following primers: *Lsd1* forward: 5′-CCAACACTAAAGAGTATCCCAAGAATA-3′; *Lsd1* reverse: 5′-GGTGATTATTATAGGTTCAGGTGTTTC-3′; *Actb* forward: 5′-AGCCAACTTTACGCCTAGCGT-3′;

*Actb* reverse: 5′-TCTCAAGATGGACCTAATACGGC-3′. The *Lsd1* reverse primer anneals to exon 6 of *Lsd1*, which is deleted in *Lsd1*$^{CAGG}$. Results were normalized to *Actb*.

**Morris water maze.** Training was carried out in a round, water-filled tub (52 inch diameter). Mice were trained with 4 trials per day for 5 days with a maximum trial length of 60 s and a 15 min intertrial interval. Subjects that did not reach the platform in the allotted time were manually guided to it. Mice were allowed 5 s on the platform to survey spatial cues. Following the 5 day training period, probe trials were performed by removing the escape platform and measuring the amount of time spent in the quadrant that originally contained the escape platform over a 60 s trial. All trials were videotaped and performance analyzed by means of MazeScan (Clever Sys, Inc.).

**Fear conditioning.** On day 1, mice were placed in a fear conditioning apparatus (Colbourn) and allowed to explore for 3 min. Following this habituation period, three conditioned stimulus-unconditioned stimulus pairings were presented with a 1 min intertrial interval. The conditioned stimulus consisted of a 20 s 85db tone and the unconditioned stimulus consisted of a 2 s foot shock that co-terminated with each conditioned stimulus. On day 2, subjects were presented with a context test by placement in day 1 conditioning apparatus and amount of freezing behavior was recorded by camera and quantified by Colbourn software. On day 3, subjects were presented with a tone test by exposure to conditioned stimulus in a novel context. Mice were allowed to explore novel context for 2 min then presented with the 85 db tone for 6 min with freezing behavior recorded.

**RNA sequencing.** Mice were anesthetized with a lethal dose of isoflurane, followed by decapitation and hippocampus dissection. Hippocampi were snap frozen with liquid nitrogen in 1 mL trizol and stored at −80 °C. For RNA isolation, samples were thawed at 37 °C then kept on ice prior to homogenization with a Polytron homogenizer with a 5 s pulse. After a 5 min incubation at room temperature, one tenth the sample volume of 1-bromo-3chloropropane was added, mixed by inversion and incubated for three minutes at room temperature. Samples were then centrifuged at 13,000×*g* for 15 min at 4 °C to separate the aqueous and organic layers. As much of the aqueous layer was recovered as possible, then RNA was precipitated with isopropanol. Pellets were then washed with 75% ethanol and resuspended in 50 µL deionized water. RNA library preparation and sequencing were performed by HudsonAlpha Genomic Services Lab. RNA was Poly(A) selected and 300 bp size selected. Libraries were sequenced for 25 million 50 bp paired-end reads.

**RNA-seq analysis.** Short read FASTQ files were quality trimmed using FASTX toolkit (v. 0.0.14) to trim three bases from the 5′ end of the reads. Paired-end reads were then mapped to the mm9 genome using tophat2[45] and the UCSC knownGene gtf file. The following parameters were used in the tophat2 call "-N 1 –g 1 –read-gap-length 1 –mate-inner-dis 170". Reads that had the same starting location and strand with mate-pairs that also had the same location and strand were considered to be PCR duplicates and removed from subsequent analyses using Picard tools (v. 1.103). Differentially expressed transcripts were determined using Cufflinks and Cuffdiff (v2.1.1)[46]. Downstream analyses were performed in R/Bioconductor[47] and used gene summarized expression levels normalized using Fragments Per Kilobase per Million (FPKM) from Cufflinks. Hierarchical clustering was performed using the pvclust R package were significance was determined using bootstrapping[48]. Principle Components Analysis (PCA) was conducted using the "prcomp" function of the stats package in R. Enriched gene ontologies were determined using the package "GOstats" (v. 3.1.1)[49]. Gene Set Enrichment Analysis (GSEA) was performed using a pre-ranked gene list determined by cuffdiff and GSEA (v. 2.1.0)[50]. Hierarchical clustering of gene expression data was performed using average clustering in the heatmap.2 package. UCSC-style display of gene expression data were plotted using the "rtracklayer" package[51] and custom R scripts to display RNA sequencing reads as histograms (available upon request)[52].

**Comparison to human gene expression data.** Normalized gene expression data from LOAD[36], FTD[37], and PD[39] patients were downloaded from Gene Expression Omnibus gene sets GSE44772, GSE13162 and GSE20295, respectively. Comparison to *Lsd1*$^{CAGG}$ gene expression data was performed by mapping mouse and human genes using the NCBI homologene database[53]. Correlation of *Lsd1*$^{CAGG}$ gene expression changes and those found in LOAD, FTD, and PD patients were assessed using Spearman's rank correlation ($\rho$). *P*-values were determined by analysis of variance (ANOVA).

**Data availability.** FastQ files for RNA sequencing experiments can be found in the GEO data set GSE98875.

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

## Acknowledgements

We thank M. Rosenfeld (U.C.S.D) for providing the LSD1 mice; N. Seyfried, R. Betarbet, M. Gearing, J. Fritz, and D. Cooper from the Emory ADRC (P50 AG025688), NINDS Emory Neuroscience Core Facilities (P30NS055077), A. Katz and S. Katz, for assistance with analyses of human tissue; J. Schroeder from the Emory Rodent Behavioral Core for help with behavioral assays; G. Pavlath for assistance with muscle histology; E. Corgiat for the myelination data; G. Bassell, V. Faundez, J. Boss, B. Kelly, C. Bean and T. Caspary for comments on the manuscript and assistance throughout; R. Tenser for introducing us to A. Levey; and all of the Katz Lab for contributions throughout. D.J.K. would like to thank J. Cohen, F. Turano, H. Lyman and S. Tilghman; D.A.M. would like to thank L. Myrick and A. Myrick; and M.A.C. would like to thank R. Cordeiro and Y. France for help along the way. A.K.E. would like to thank A. Wiemer and P. Engstrom. D.A.M. was supported by the Emory PREP Post-Bac Program (5R25GM089615-04); and M.A.C. and B.G.B. by the GMB training grant (T32GM008490-21). B.G.B. was supported by NIH

pre-doctoral fellowship F31AI11226101. A.K.E. was supported by the BCDB training grant (T32GM008367-26) D.W. was supported by AG0476678 and KPS was supported by NS098615. The work was supported by a grant to D.J.K. from the National Institute of Neurological Disorders and Stroke (1R01NS087142).

## Author contributions

M.A.C. and D.A.M. contributed equally to this work. D.J.K. worked on the design and execution of the experiments as well as the writing of the manuscript. A.K.E., K.A.P.-S., and D.W. assisted with experiments. B.G.B. and J.M.B. contributed to all of the bioinformatic analyses. A.I.L. assisted with obtaining and analyzing the human pathology and provided guidance throughout. The manuscript was edited by all the coauthors.

## Additional information

**Competing interests:** The authors declare no competing financial interests.

