## [Peer Review File · Nature Communications]

Reviewers' comments:

Reviewer #1 (Remarks to the Author):

In this study, Christopher et al. generated conditional mutant mice, by deleting histone demethylase LSD1 in adult mice, and found that a general loss of LSD1 results in hippocampal and cortical neurodegeneration. The authors crossed the Cagg-CreER mice with LSD1^{floxP} mice and generated LSD1^{CAGG} mice. A general deletion of LSD1 in adult mice caused widespread neuronal death in the hippocampus and cortex. The mutant mice exhibited severe motor deficit and memory impairment as well as increased stem cell gene expression in the hippocampus. Based on the RNA-seq analysis, the authors suggested that those genes associated with neuroinflammation and neurodegeneration were affected in the hippocampus of the mutant mice. Finally, they demonstrated that LSD1 protein was mislocalized in the brains of AD and FTD patients.

The major shortfall of this study is the lack of detailed characterization of the conditional Lsd1^{CAGG} mutant mice. Only one citation for the Cagg-Cre mice was provided without further description; the expression profile of LSD1 in the conditional mutant mice after tamoxifen injection was not included. Given that Cre recombinase was induced in a wide range of cells and organs in the Cagg-Cre mice by tamoxifen, it is difficult to analyze whether LSD1 in neurons or in the brain exert the protective role against neurodegeneration, without knowing where and when the loss of LSD1 was induced in the mice. In particular, since severe defects including paralysis and muscle atrophy were observed in the conditional mutant mice at the terminal stage, it is possible that neurodegeneration was a secondary effect of those defects. The global transcriptional change of hippocampus in LSD1^{CAGG} mutant mice was evaluated at the terminal stage, at which a vast amount of neuronal cell death occurred. The authors suggested that the up-regulation of specific genes in LSD1^{CAGG} mice pointed towards a role of LSD1 as a transcriptional repressor. However, there is no concrete experimental evidence supporting that the up-regulated genes are the targets of LSD1. The change of the different genes (including those related to neurodegeneration and neuroinflammation) in the hippocampus of the conditional mutant mice at the terminal phenotype may just reflect the neurodegeneration status of the mice. Finally, more experimental evidence is required to support that LSD1 in neurons was mislocalized during neuronal degeneration upon the progression of various diseases including AD and FTD.

Specific points:

Detailed descriptions (including source and citations) of Cagg-Cre and floxed Lsd1 mice need to be included.

The period of tamoxifen injection for different analyses needs to be stated. Also, the authors need to define the terminal stage, at which most of the analyses were conducted.

The authors showed that expression of LSD1 was absent in the dying cells of hippocampus and cortex of LSD1^{CAGG} mutant mice at the terminal stage (Fig. 1X-aa). However, prominent expression of LSD1 was observed in the non-pyknotic nuclei of hippocampus one week before the terminal motor phenotype (Suppl. Fig. 4h, i). Additional experiments are required to demonstrate that the loss of LSD1 in the hippocampus or cortex is responsible for the neurodegeneration phenotype.

To evaluate the memory performance of the mutant mice by MWM, the motor activity of the mutant mice after tamoxifen injection (28 days injection + 5 day MWM test) needs to be shown. Since severe neurodegeneration was observed in the mutant mice, their visual ability (e.g. the ability to locate a visible platform) needs to be evaluated.

The results of FC do not seem to be statistically significant (Fig. 2e).

Fig. 5: The genome-wide correlation of the hippocampal gene expression data of LSD1CAGG mutant mice with the AD, FTD, and PD cases provide very little information. For example, if no filtering criteria is included, the differential expression of tissue-specific genes may interfere with the results (as the data were compared between hippocampus and different brain tissues including cortex, cerebellum and substantia nigra).

Reviewer #2 (Remarks to the Author):

The manuscript by Christopher et al. provides evidence linking histone demethylase LSD1 to neurodegeneration. Using an inducible whole-body deletion of LSD1 (*Lsd1CAGG*) in adult mice, the authors demonstrate that LSD1 loss leads to tissue and region specific neuronal cell death in the hippocampus and cortex of adult mice, as well as learning and memory deficits and paralysis. Additionally, they observe transcriptional changes in neurodegeneration pathways common to human neurodegenerative diseases such as AD and FTD, as well as reactivation of stem cell genes in the hippocampus.

This is one of the first studies addressing the *in vivo* function of LSD1 in differentiated cells and specifically in the adult brain, and the first study to link its function to neurodegeneration. However, while the authors provide relatively thorough characterization of the effect of LSD1 loss in the adult hippocampus, they fail to tease apart the function of LSD1 in specific cell types within the hippocampus. Provided the following points are properly addressed, we believe the manuscript will make a fine contribution to your journal:

1. Based on the RNA-seq analysis, Christopher et al show upregulation of inflammatory response and complement cascade pathways suggesting dysregulation of microglial function in the *Lsd1CAGG* mice. They also observe strong gliosis response (Fig. 1p-s). To properly evaluate whether the effect of *Lsd1CAGG* is cell autonomous, and dissect the contributions of individual cell types, the authors should provide additional information about the cell type specific expression pattern of LSD1 in the hippocampus. Immunofluorescence co-staining with cell type specific markers such as GFAP and Iba1 should be included. The same type of analysis should be extended to some of the stem cell genes that appear to be upregulated in the *Lsd1CAGG* mice.

2. Along the same lines, it would be interesting to tease apart the contributions of individual pathways to neuronal cell death. To that end, the authors should examine the correlation of gene expression changes of individual pathways with those in various human neurodegenerative disorders (Fig 5).

3. The authors claim that impaired performance in the water maze test is not due to motor deficits because they did not observe a difference in swimming speeds in the *Lsd1CAGG* mice relative to littermate controls. Supplementary Fig 6a does not provide strong support for this observation. In the figure legend authors should include information about the statistical test used. Additionally it would be useful to perform open field test as an additional assessment of locomotor activity.

4. It would be interesting to examine whether motor defects can be linked to glial (Shwann cell) or neuronal cell death in the motor cortex.

Minor points:

1. Minor reorganization of some of the panels might be useful to make the study easier to follow.
 - a. In Figure 1, LSD1 staining should be shown early on, probably immediately after figure 1e.
 - b. In Figure 5, panels e and f, should be moved below the corresponding panels for the cortex in the same diseases, i.e. panels a and b.

2. In Figure 6b the authors show percent colocalization with pTau in only 5 FTD-TDP43 patients, but in the text (page 16, line 341) it appears they examined 14 cases. Were patients excluded from the analysis and why? Also, the y-axis should specifically state Percent Colocalization of LSD1 with pTau.

Main Reviewer Comment:

The central critique of this work centers around the original experimental setup.

Reviewer 1 writes that “the major shortfall of this study is the lack of detailed characterization of the conditional *Lsd1*^{CAGG} mutant mice. Given that Cre recombinase was induced in a wide range of cells and organs in the Cagg-Cre mice by tamoxifen, it is difficult to analyze whether LSD1 in neurons or in the brain exert the protective role against neurodegeneration... In particular, since severe defects including paralysis and muscle atrophy were observed in the conditional mutant mice at the terminal stage, it is possible that neurodegeneration was a secondary effect of those defects.”

Reviewer 2 focuses this critique by saying “However, while the authors provide relatively thorough characterization of the effect of LSD1 loss in the adult hippocampus, they fail to tease apart the function of LSD1 in specific cell types within the hippocampus.”

Reviewer 1 does not give any guidance as to how to address this issue. However, reviewer 2 suggests an experimental approach to begin to tease this apart. “Based on the RNA-seq analysis, Christopher et al show upregulation of inflammatory response and complement cascade pathways suggesting dysregulation of microglial function in the *Lsd1*^{CAGG} mice. They also observe strong gliosis response (Fig. 1p-s). To properly evaluate whether the effect of *Lsd1*^{CAGG} is cell autonomous, and dissect the contributions of individual cell types, the authors should provide additional information about the cell type specific expression pattern of LSD1 in the hippocampus. Immunofluorescence co-staining with cell type specific markers such as GFAP and Iba1 should be included. The same type of analysis should be extended to some of the stem cell genes that appear to be upregulated in the *Lsd1*^{CAGG} mice.”

Author Response: To address the function of LSD1 in terminally differentiated hippocampal neurons, *in vivo*, in the mammalian brain, we originally considered two potential experimental approaches. The first approach would be to inducibly delete *Lsd1* throughout the adult mouse. A second approach would be to experimentally delete *Lsd1* only in hippocampal neurons.

Reviewer 1 commented that the neurodegeneration in *LSD1*^{CAGG} mutants could be due to secondary effects. While this is certainly true, this may also be true in the human cases. Thus, we chose the first approach because it is more physiologically relevant to human neurodegeneration. For example, this approach allows us to make the central conclusion of the paper, that LSD1 protects against hippocampal and cortical neurodegeneration in a manner that is potentially related to Alzheimer’s Disease and Frontotemporal Dementia. Reviewer 2 clearly acknowledges this, writing that “This is one of the first studies addressing the *in vivo* function of LSD1 in differentiated cells and specifically in the adult brain, and the first study to link its function to neurodegeneration.”

A limitation of our approach is that it does not enable us to determine whether the requirement for LSD1 is autonomous versus non-autonomous, or potentially whether the neurodegeneration could be due to a secondary effect resulting from another brain region or tissue. Knowing the specific cell type requirement might improve the work, especially for a neuroscience audience, but it doesn't preclude our ability to make the central conclusion of the paper. As a result, we readily acknowledged this limitation of our approach in the initial submission (lines 131-135). Moreover, it is not clear if there is a definitive way to address this critique. The most definitive way to address this would be to inducibly delete *Lsd1* with an inducible hippocampus neuron-specific *Cre*. Unfortunately, at the moment no such *Cre* line exists. However, as stated above, reviewer 2 suggests a thoughtful approach to begin to tease this apart. We believe that by carrying out the co-staining experiments suggested by reviewer 2, we will be able to strengthen the case that the neurodegeneration is likely cell autonomous.

Reviewer #1 (Remarks to the Author):

Reviewer Comment: Detailed descriptions (including source and citations) of Cagg-Cre and floxed *Lsd1* mice need to be included.

Author Response: In the acknowledgements we thanked M. Rosenfeld for providing the published floxed mice. However, we neglected to cite the original reference to the floxed allele in the results when we first mention it. We are very sorry for this major oversight. We have now referenced the original publication, as well as several other publications that have utilized the *Lsd1* floxed allele (line 79). We have also added additional citations for the use of the well characterized CAGG-Cre line (line 80).

Reviewer Comment: the expression profile of LSD1 in the conditional mutant mice after tamoxifen injection was not included. Given that Cre recombinase was induced in a wide range of cells and organs in the Cagg-Cre mice by tamoxifen, it is difficult to analyze whether LSD1 in neurons or in the brain exert the protective role against neurodegeneration, without knowing where and when the loss of LSD1 was induced in the mice.

Author Response: Because the data is contained within several figures, perhaps the reviewer is not aware that we have included the expression profile of LSD1 after tamoxifen injection in the CA1 (Figure 1x,y), cortex (Figure 1z, aa), cerebellum (Supplemental Figure 4e,f), kidney (Supplemental Figure 5c,f) and liver (Supplemental Figure 5 i,l). We have also quantified the extent of deleted *Lsd1* allele in the hippocampus, cerebellum and cortex (Supplemental Figure 4g). We believe that this is a thorough characterization, but we are happy to provide additional characterization if necessary.

Reviewer Comment: The global transcriptional change of hippocampus in LSD1CAGG mutant mice was evaluated at the terminal stage, at which a vast amount of neuronal cell death occurred.

Author Response: The reviewer is correct. However, this is also true of the human cases to which we compared our expression changes, so we believe that this is the appropriate stage for comparison.

Reviewer Comment: The authors suggested that the up-regulation of specific genes in LSD1CAGG mice pointed towards a role of LSD1 as a transcriptional repressor.

Author Response: We agree with reviewer 1. This speculative statement has been removed from the text (line 216).

Reviewer Comment: The change of the different genes (including those related to neurodegeneration and neuroinflammation) in the hippocampus of the conditional mutant mice at the terminal phenotype may just reflect the neurodegeneration status of the mice.

Author Response: Reviewer 1 brings up an important point. We too strongly considered this possibility. For this reason, we included a control comparison between our RNA-seq data set and a human Parkinson Disease substantia nigra expression data set. The fact that there is far less overlap between our expression changes and the Parkinson's Disease expression changes, despite the substantial neurodegeneration in the substantia nigra in these cases, suggests that the major overlap in expression changes that we see with AD and FTD cases does not just reflect the neurodegeneration status. This argument is also true for the sporadic FTD cases, which show much less overlap than FTD-progranulin cases, despite the identical extent of neurodegeneration.

As reviewer 2 has suggested (see below), we are also happy to further compare to additional neurodegenerative disorders. We believe that these additional comparisons will further strengthen our case. Moreover, even if the expression changes were to just reflect the neurodegeneration status, this may also be true in the human cases, so the comparison (and substantial overlap that we observe) to the human cases would still be valid.

Reviewer Comment: Finally, more experimental evidence is required to support that LSD1 in neurons was mislocalized during neuronal degeneration upon the progression of various diseases including AD and FTD

Author Response: We tried to carefully avoid conclusions that LSD1 mislocalization is linked to the "progression" of neurodegenerative disease, because it is very well established in the field that neuropathological features of neurodegenerative disease do not necessarily track with progression. A glaring example is amyloid plaques, which accumulate in pre symptomatic phases of Alzheimer's disease, rather than tracking with disease progression. Neurofibrillary tangles, the other major hallmark of Alzheimer's disease, also correlate with disease progression less strongly than synapse loss. Obviously, the lack of correlation with disease progression does not minimize their importance as central features of Alzheimer's Disease. Rather, similar to what we are proposing for LSD1, these pathologies are likely part of a cascade of pathophysiological processes. Indeed, unlike amyloid and neurofibrillary tangles, which have little or no impact on neurodegeneration in mouse models, our discovery of LSD1 mislocalization is stronger because of our demonstration that striking neurodegeneration results from the conditional knockout of LSD1 in mice.

Reviewer Comment: The period of tamoxifen injection for different analyses needs to be stated.

Author Response: We are unclear what the reviewer is referring to. The period of tamoxifen injection was detailed (lines 457-459) in the methods and is the same in all experiments performed.

Reviewer Comment: Also, the authors need to define the terminal stage, at which most of the analyses were conducted.

Author Response: The terminal phenotype was shown in Supplementary Video 1 and described in the supplemental figure legend. To make this clearer, we have now also included an additional reference to this terminal phenotype in the text (lines 85-88).

Reviewer Comment: To evaluate the memory performance of the mutant mice by MWM, the motor activity of the mutant mice after tamoxifen injection (28 days injection + 5 day MWM test) needs to be shown.

Author Response: The swimming speed and distance covered was included in supplemental figure 6. Although the speed travelled trended towards less in the LSD1^{CAGG} mutants, this is likely due to the fact that the mutant mice occasionally pause from swimming and stay in place. It is unclear why the LSD1^{CAGG} mutants occasionally do this. However, the defect in latency is clearly not due to the inability of LSD1^{CAGG} mutants to swim. This can be seen by the fact the mutants cover more distance on day 5 (supplemental figure 6B), when they show a defect in locating the platform. To further illustrate this point, we have now included a representative video of control and mutant mice performing the Morris Water Maze on the day the defect was observed (Supplementary Video 2). The text has also been amended to make this clearer (lines 181-186).

Reviewer Comment: Since severe neurodegeneration was observed in the mutant mice, their visual ability (e.g. the ability to locate a visible platform) needs to be evaluated.

Author Response: Although we did not detect any evidence of visual impairment, the reviewer correctly points out that we cannot rule out the contribution of a slight visual impairment. We have added text to the results to specifically acknowledge this possibility (lines 196-199).

Reviewer Comment: The results of FC do not seem to be statistically significant (Fig. 2e).

Author Response: The reviewer correctly points out that the contextual fear conditioning assay is not statistically significant when compared by repeated measures two-way ANOVA across all time points. Nevertheless, we observe less time spent freezing at every time point assayed (Figure 2C). Especially when compared to the cued fear conditioning assay (Figure 2D), that is indistinguishable between controls and LSD1^{CAGG} mutants, it is clear that there is a defect in contextual fear conditioning. In addition, we have now noted the individual time points that are statistically significantly between controls and LSD1^{CAGG} mutants when compared by an unpaired t-test (lines 191-193, Figure 2c).

Reviewer Comment: Fig. 5: The genome-wide correlation of the hippocampal gene expression data of LSD1CAGG mutant mice with the AD, FTD, and PD cases provide very little information. For example, if no filtering criteria is included, the differential expression of tissue-specific genes may interfere with the results (as the data were compared between hippocampus and different brain tissues including cortex, cerebellum and substantia nigra).

Author Response: The reviewer is correct in noting that different tissues were compared and no filtering criteria were used. As a result, the differences in tissue specific expression should bias the results against seeing a genome-wide overlap in expression. The fact that we still observe a substantial overlap in genome-wide expression changes, despite this, strongly suggests that this overlap is real.

Reviewer #2 (Remarks to the Author):

Reviewer Comment: Along the same lines, it would be interesting to tease apart the contributions of individual pathways to neuronal cell death. To that end, the authors should examine the correlation of gene expression changes of individual pathways with those in various human neurodegenerative disorders (Fig 5).

Author Response: We thank the reviewer for this thoughtful comment. As stated above, we would be happy to further compare our gene expression changes to changes observed in additional neurodegenerative disorders.

Reviewer Comment: The authors claim that impaired performance in the water maze test is not due to motor deficits because they did not observe a difference in swimming speeds in the Lsd1CAGG mice relative to littermate controls. Supplementary Fig 6a does not provide strong support for this observation. In the figure legend authors should include information about the statistical test used. Additionally it would be useful to perform open field test as an additional assessment of locomotor activity.

Author Response: The detailed response is included above. In addition, as suggested, we have added the statistical test used (Supplemental Figure legend 6).

Reviewer Comment: It would be interesting to examine whether motor defects can be linked to glial (Shwann cell) or neuronal cell death in the motor cortex.

Author Response: We too strongly considered these possibilities. We have specifically examined myelination in the spinal cord and neuronal cell death in the motor cortex. Unfortunately, we did not observe any myelination defect, by MBP staining, in the spinal cord. This is now included as Supplementary Figure 1e,f (lines 99-101). In addition, careful examination revealed a large amount of neuronal cell death in the motor cortex. In the initial submission, we chose not to specifically include this in the paper because it is not possible at this moment to determine if this defect is responsible for the observed paralysis. We have now specifically included a statement to this effect (lines 118-120 and 124-126) and added the data to Supplementary Figure 2q,r.

Minor points:

1. Minor reorganization of some of the panels might be useful to make the study easier to follow.
 - a. In Figure 1, LSD1 staining should be shown early on, probably immediately after figure 1e.
 - b. In Figure 5, panels e and f, should be moved below the corresponding panels for the cortex in the same diseases, i.e. panels a and b.

Author Response: In Figure 1, the LSD1 staining has been placed at the end of the figure to match the order and logic of the text. In Figure 5, we have now moved the panels. We thank the reviewer for this thoughtful suggestion.

Reviewer Comment: In Figure 6b the authors show percent colocalization with pTau in only 5 FTD-TDP43 patients, but in the text (page 16, line 341) it appears they examined 14 cases. Were patients excluded from the analysis and why? Also, the y-axis should specifically state Percent Colocalization of LSD1 with pTau.

Author Response: No cases were excluded from the analysis. The LSD1 IHC worked on all 14 AD cases and all 14 FTD cases. In addition, the co-localization by dual immunofluorescence worked well on all 14 AD cases, but only worked well on 5 FTD cases. In the other cases, the dual immunofluorescence was not good enough to quantify the co-localization. This is why only 5 FTD cases could be quantitated for co-localization.

Reviewers' comments:

Reviewer #1 (Remarks to the Author):

In the revised manuscript, while the authors addressed most of the concerns raised, the most critical concern, i.e. dissecting the functions of LSD1 in specific cell types in neurodegeneration, was not strongly supported by the newly added data. In particular, it is unclear to the reviewer which data in the rescue experiment demonstrated that the re-expression of LSD1 in hippocampal neurons eliminated the hippocampal neuronal cell death in the transgenic mice (Supplementary Figure 4g-l). The authors did not indicate what parameter was used to measure the hippocampal neuronal death in the text nor in the Figure legend. Also, the data of the Lsd1CAGG mice without pAAV6-Syn-Lsd1 expression was not included as the control for comparison with the transgenic mice with the virus overexpression.

The other experimental evidence supporting the conclusion that there is a cell autonomous requirement for LSD1 in hippocampal neurons was only correlative (page 1 of the response letter). Specifically, the newly added data in Supplementary Fig. 4 was incorrectly labelled as Supplementary Figure 12 in the response letter (page 1).

Reviewer #2 (Remarks to the Author):

The authors have made several improvements to the manuscript in response to our suggestions. They have performed additional experiments showing that there is a cell autonomous requirement for LSD1 for neuronal survival, including a rescue experiment which shows that cell-type specific expression of LSD1 from a neuron-specific promoter/virus is sufficient to prevent neuronal cell death. These experiments don't necessarily speak to the actual contributions of the other brain cell types in the neuronal cell death phenotype triggered by the LSD1 loss, i.e. whether cell-type specific LSD1 loss is sufficient to induce cell death, however they do confirm that neuronal LSD1 is required for the process. In that respect, it would have been informative to have data regarding the status of LSD1 expression in microglia, especially as the RNA-seq results show significant upregulation of inflammatory response and complement cascade pathways, which indicates a strong microglia contribution to the phenotype. Indeed, co-staining with microglial markers (such as Iba1, for which effective IHC antibody raised in guinea pig is in fact available, Synaptic Systems #234004) would have been an easy way to tease apart individual cell-type specific effects and contributions, and at least to some degree, circumvent the limitations of the KO approach used, as well as the lack of cell-type specific RNA-seq. Nevertheless, despite lacking mechanistic details and the limitations of the approach, the manuscript does provide a novel function for LSD1 in neurodegeneration and a thorough characterization of the effect of LSD1 loss.

We have provided our detailed response to the reviewers in *italics*, interspersed between the reviewers comments in **bold**.

Reviewers' comments:

Reviewer #1 (Remarks to the Author):

In the revised manuscript, while the authors addressed most of the concerns raised, the most critical concern, i.e. dissecting the functions of LSD1 in specific cell types in neurodegeneration, was not strongly supported by the newly added data.

Reviewer 2 also pointed out that the experiments in our previously revised manuscript don't necessarily speak to the actual contributions of the other brain cell types in the neuronal cell death phenotype triggered by the LSD1 loss, i.e. whether cell-type specific LSD1 loss is sufficient to induce cell death

In our previously revised version, we showed that LSD1 is present in astrocytes, but LSD1 protein levels are not changed in astrocytes when our Lsd1 mutants reach the terminal phenotype. However, we were unable to assay LSD1 in microglia because we were unaware of a microglia antibody that was not raised in rabbit. We thank reviewer 2 for bringing the guinea pig IBA1 antibody to our attention. To our surprise, we have found that LSD1 is not expressed in microglia. In addition, we have gone on to show that although LSD1 is expressed oligodendrocytes, LSD1 protein levels are also not changed in oligodendrocytes when our Lsd1 mutants reach the terminal phenotype. As a result, we now realize that, within the brain, our inducible mouse model results in LSD1 protein loss only in neurons. This data is now included as Supplementary Fig. 1-4 (along with Fig. 1a-d).

This data showing neuron-specific LSD1 loss substantially alters the interpretation of our experiments, by allowing us to conclude that loss of LSD1 specifically in neurons is sufficient to induce the observed widespread neurodegeneration in our mice. We have significantly altered the introduction, results and discussion sections of the manuscript to reflect this conclusion. In

addition, this conclusion is consistent with our high/low tamoxifen mosaic experiments, and viral rescue experiments (Supplementary Fig. 7), which indicate that LSD1 acts cell autonomously in hippocampal neurons.

In particular, it is unclear to the reviewer which data in the rescue experiment demonstrated that the re-expression of LSD1 in hippocampal neurons eliminated the hippocampal neuronal cell death in the transgenic mice (Supplementary Figure 4g-l). The authors did not indicate what parameter was used to measure the hippocampal neuronal death in the text nor in the Figure legend.

We appreciate the reviewer's point about the need to further characterize the viral rescue. In addition to the histology, we have now examined dendrites and axons in the rescued mice. We have also assayed for apoptosis and quantified total hippocampal nuclei/pyknotic hippocampal nuclei. This new data is included in Supplementary Fig. 7. Taken together, we find no evidence of neuronal cell death, or any other morphological changes, in the Lsd1 inducibly deleted mice rescued by Lsd1 neuronal specific virus.

Also, the data of the Lsd1CAGG mice without pAAV6-Syn-Lsd1 expression was not included as the control for comparison with the transgenic mice with the virus overexpression.

The representative mouse shown in Supplemental Fig. 7 showed no evidence of any phenotype 14 weeks post tamoxifen injection, well beyond the time when all previous uninjected controls had died. This is in contrast to the uninjected controls, which showed the expected widespread distribution of pyknotic nuclei throughout the hippocampus and cortex. We originally chose not to include data from uninjected controls because they die many weeks before the rescued mice, so they were no longer age-matched. In the newly revised version we have included these uninjected controls and denoted the difference in age in the figure legend (Supplementary Figure 7).

The other experimental evidence supporting the conclusion that there is a cell autonomous requirement for LSD1 in hippocampal neurons was only correlative (page 1 of the response letter).

The mosaic analysis in Supplemental Figure 7 is correlative. However this is true of all mosaic analysis, which is the standard in the field for defining cell autonomy. The correlation at both high and low tamoxifen doses argue that the neurodegeneration induced by LSD1 deficiency is cell autonomous. If astrocytes or microglia were somehow causing the neuronal cell death in our mice, we would not expect to see LSD1 protein lost only in the neurons that died. In addition, this conclusion is now strengthened by the finding that LSD1 is not expressed in microglia, and not compromised in astrocytes or oligodendrocytes at the terminal phenotype (Supplementary Fig. 2-4).

Specifically, the newly added data in Supplementary Fig. 4 was incorrectly labelled as Supplementary Figure 12 in the response letter (page 1). We sincerely regret any inconvenience this oversight in proofreading caused.

Reviewer #2 (Remarks to the Author):

The authors have made several improvements to the manuscript in response to our suggestions. They have performed additional experiments showing that there is a cell autonomous requirement for LSD1 for neuronal survival, including a rescue experiment which shows that cell-type specific expression of LSD1 from a neuron-specific promoter/virus is sufficient to prevent neuronal cell death. These experiments don't necessarily speak to the actual contributions of the other brain cell types in the neuronal cell death phenotype triggered by the LSD1 loss, i.e. whether cell-type specific LSD1 loss is sufficient to induce cell death (*please see our new data presented above*), however they do confirm that neuronal LSD1 is required for the process. In that respect, it would have been informative to have data regarding the status of LSD1 expression in microglia, especially as the RNA-seq results show significant upregulation of inflammatory response and complement cascade pathways, which indicates a strong microglia contribution to the phenotype. Indeed, co-staining with microglial markers (such as Iba1, for which effective IHC antibody raised in guinea pig is in fact available, Synaptic Systems #234004) would have been an easy way to tease apart individual cell-type specific effects and contributions, and at least to some degree, circumvent the limitations of the KO approach used, as well as the lack of cell-type specific RNA-seq. We are grateful for the thoughtful suggestion, which substantially changed the interpretation of our experiments. As detailed above, this experiment led us to conclude that LSD1 expression is only compromised in neurons (Fig 1 a-d and Supplementary Fig. 1-4).

Nevertheless, despite lacking mechanistic details and the limitations of the approach, the manuscript does provide a novel function for LSD1 in neurodegeneration and a thorough characterization of the effect of LSD1 loss. We appreciate the reviewer's comment about the novel advance that our work provides. We hope that the reviewers agree that the new data presented support our conclusion that loss of LSD1 specifically in neurons is sufficient to induce the widespread neurodegeneration observed in the brains of our mice. Combined with the positive aspects of our work, consistently noted by the reviewers throughout the review process, we hope that the manuscript will now be suitable for publication in Nature Communications.

REVIEWERS' COMMENTS:

Reviewer #2 (Remarks to the Author):

Christopher et al provide additional characterization of the Lsd1CAGG mice further dissecting the cell type-specific function of LSD1 in the adult brain, showing that LSD1 is not expressed in microglia, and not compromised in astrocytes or oligodendrocytes at the terminal phenotypes. These results substantially strengthen the interpretation of the experiments suggesting that indeed LSD1 expression is only compromised in neurons and is sufficient to induce neurodegeneration in the Lsd1CAGG mice. We recommend this manuscript for publication.